# LRConfNet: Logical Reasoning-Driven Confidence Adjustment and Regularization for Hierarchical Classification of Degraded Images

## Abstract

Hierarchical classification (HC) is widely applied in remote sensing and natural image analysis. However, real-world degradations—such as noise, blur, occlusion, and low resolution—often compromise fine-grained predictions for HC. Existing methods struggle to balance coarse- and fine-level accuracy, handle sparse hierarchies, and integrate multi-modal features, particularly under low-confidence predictions and complex semantic structures. We propose LRConfNet, a unified framework that addresses these challenges by combining Uncertainty Quantification (UQ) with Logical Reasoning Regularization (LogReg) to dynamically adjust classification paths. A Vision Transformer backbone extracts global visual features, while a Semantic-Guided Cross-Attention module enables multi-modal fusion. When fine-grained confidence is low, LRConfNet triggers a logic-driven hierarchical fallback mechanism, guided by LogReg, to back off to coarse-level predictions and avoid over-classification. To further enhance generalization, we introduce a multi-level loss optimization strategy with adaptive weight adjustment. An attention enhancement loss and attention-gradient fusion are incorporated to refine spatial focus, especially confronting degraded conditions and data scarcity. Moreover, a position prompting mechanism reinforces feature selection in sparse hierarchies. Extensive experiments on degraded remote sensing and natural image benchmarks show that LRConfNet significantly outperforms SOTA methods, demonstrating superior robustness and adaptability.

## 1 Introduction

With the growing deployment of computer vision systems in domains such as remote sensing and natural image analysis, image classification is facing increasingly complex challenges. Hierarchical Classification (HC), which leverages semantic taxonomies to organize labels across multiple granularity levels (e.g., order → family → species), has been widely applied in diverse fields, including remote sensing, text classification, and medical diagnosis (Chen et al., 2022; Li et al., 2022; Zeng et al., 2022; Novack et al., 2023; Wang et al., 2024; Kowsari et al., 2020; Kulbhushan et al., 2023; Li et al., 2023; Xiong et al., 2023). Unlike flat classification, HC enables progressive refinement of predictions by capturing semantic dependencies between coarse and fine labels (Yuan et al., 2024; Miranda et al., 2023).

HC has proven effective in various real-world tasks, particularly where label structures are inherently hierarchical, such as ship recognition in aerial imagery or multi-level sentiment analysis in texts. However, real-world degradations—such as noise, occlusion, blur, and low resolution—as well as class imbalance, long-tail distribution, and sample sparsity in HC, severely hinder the extraction of discriminative features required for fine-grained classification (Yazici et al., 2022). These issues lead to the emergence of low-confidence samples and exacerbate semantic sparsity across hierarchy levels, resulting in severe error propagation that undermines model robustness and accuracy.

Most current HC methods focus on top-down inference from coarse to fine levels but lack mechanisms to dynamically adjust classification confidence. In particular, they rarely include confidence-aware fallback strategies, which are essential for mitigating misclassification propagation under low-confidence conditions (Yazici et al., 2022).

Moreover, current multi-modal fusion techniques and hierarchical loss designs exhibit limited capacity to balance performance across hierarchical levels, thereby restricting the overall efficacy (Li et al., 2022; Zeng et al., 2022). These limitations are especially pronounced when dealing with low-quality or sparsely labeled data, illustrated in figure 1. As semantic cues become less distinguishable, uncertainty increases and hierarchical fallback becomes essential for reliable classification.

To address these challenges, we propose LRConfNet, a dynamic confidence-aware HC framework that integrates Uncertainty Quantification (UQ) and Logical Reasoning Regularization (LogReg). Our goal is to improve HC under degraded input conditions and data scarcity. The key contributions of this paper are as follows:

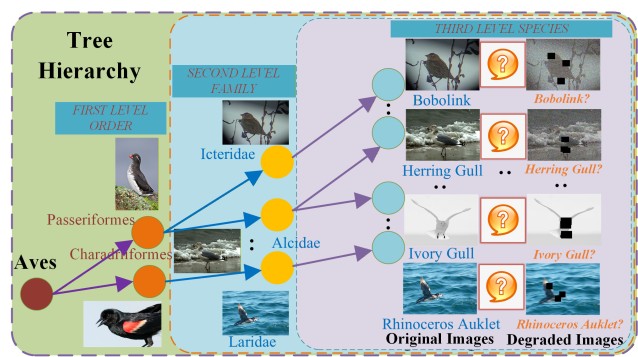

Figure 1: Hierarchical Prediction under Degradation: Visualizing Semantic Ambiguity in Tree-Structured Classification. This figure illustrates a three-level bird taxonomy—Order, Family, and Species—with semantic parent-child relations. Each species node is accompanied by representative samples under both clean and degraded conditions. The question mark icons indicate increased prediction uncertainty due to degradation.

- **A dynamic confidence adjustment framework based on UQ and LogReg.** We employ a Vision Transformer backbone to extract global features, and integrate them with semantic priors using a Semantic-Guided Cross-Attention module for multi-modal fusion. UQ is used to evaluate prediction confidence dynamically, and LogReg encourages semantically consistent predictions by regulating classification paths. This enables the model to adaptively refine decisions, especially under class imbalance and hierarchical sparsity, ensuring reliable outcomes under degraded conditions and rare-class scenarios.

- **A logic-driven hierarchical fallback optimization mechanism.** When fine-grained confidence is low, our framework triggers a semantically guided fallback strategy to coarser levels. This mechanism is driven by LogReg to enhance feature selection and reasoning during fallback, allowing the model to focus on critical regions and suppress over-classification errors. The strategy not only mitigates the effects of hierarchical sparsity but also improves robustness in both degraded and long-tail sample conditions.

- **A multi-level loss optimization strategy with Position-Aware Attention Enhancement (PAAE) and position prompting.** We design a unified loss that dynamically adjusts weights across hierarchy levels to balance coarse- and fine-grained learning. In addition, an attention enhancement loss and an attention-gradient fusion mechanism refine the model's ability to focus on informative regions, especially under uncertainty. We further introduce a position prompting strategy to strengthen attention localization and hierarchical feature selection in structurally sparse scenarios, ultimately improving classification precision and stability.

To evaluate the effectiveness of LRConfNet, we conduct experiments on remote sensing and natural image datasets corrupted by varying degrees of degradation, demonstrating superior robustness and accuracy, particularly under uncertain and complex conditions.

## 2 RELATED WORK

### 2.1 HIERARCHICAL CLASSIFICATION

HC exploits the hierarchical relationships between labels to categorize data points across multiple levels. It has found broad applications in various fields, including text classification (Lefebvre et al.,

2024), functional genomics (Jiang et al., 2023; Caron et al., 2024; Yang et al., 2023), and image classification (Yuan et al., 2024). In the context of remote sensing, HC has demonstrated its effectiveness in tasks such as land cover classification (Zhao et al., 2024; Tian et al., 2024; Miao et al., 2023) and hyperspectral image analysis (Cao et al., 2024; Song et al., 2024; Sheng et al., 2024; Shi et al., 2023). Unlike flat classification, HC models capture label dependencies through tree structures or directed acyclic graphs (DAGs), facilitating semantic consistency and enabling information flow across hierarchical levels (Chen & Qian, 2021).

Recent advancements in HC have focused on improving model performance through architectural innovations and the development of custom loss functions. For instance, HMC-LMLP (Cerri et al., 2016) assigns a dedicated classifier for each hierarchy level, while HMCN (Wehrmann et al., 2018) captures inter-level relationships via local output layers. Additionally, models like C-HMCNN (Giunchiglia & Lukasiewicz, 2020) introduce parent-child consistency constraints to reinforce hierarchical coherence. Despite these advances, most methods lack confidence-aware control and rely on static thresholds, limiting adaptability to hierarchical uncertainty and under-represented branches typical of remote sensing. Beyond the above methods, several very recent works further explore hierarchical reasoning with large models or external category guidance and visually consistent HC on standard (non-degraded) images (Tan et al., 2025; Zhao et al., 2025; Park et al., 2025), as well as hierarchical selective classification and rejection and selective prediction with abstention in flat label spaces (Goren et al., 2024; Rabanser & Papernot, 2025). However, these approaches typically operate on standard (non-degraded) imagery and optimize fixed or post-hoc confidence rules to trade off accuracy and coverage, whereas our LRConfNet specifically targets degraded hierarchical image classification and jointly integrates feature-complexity–based uncertainty, logical reasoning regularization, and hierarchical fallback into a unified framework.

### 2.2 VISION TRANSFORMERS FOR HIERARCHICAL CLASSIFICATION

Vision Transformers (ViTs) have achieved strong performance on a wide range of vision tasks, including image classification and fine-grained recognition, by modeling long-range dependencies through self-attention over patch tokens rather than local convolutional windows (Dosovitskiy et al., 2020; Liu et al., 2021; Vasu et al., 2023; Carion et al., 2020). This global context is well suited to hierarchical classification (HC), where predictions across multiple semantic levels should remain consistent. Recent works explore ViT-based architectures for HC and fine-grained recognition, typically augmenting the backbone with semantic-guided attention or hierarchy-aware heads to better align visual features with label structures (Wang et al., 2023b), but they seldom address (i) uncertainty-aware, dynamic confidence adjustment under degradation, (ii) logic-driven hierarchical fallback to avoid over-classification when fine-level predictions are unreliable, and (iii) joint optimization of confidence calibration and hierarchical consistency within a unified framework.

In this work, we adopt a ViT backbone as the visual encoder and build LRConfNet on top of its global and patch-level representations. ViT features are refined by a semantic-guided cross-attention module to obtain level-specific embeddings, which are then used by LogReg for hierarchical consistency regularization and feature-complexity-based uncertainty quantification that drives dynamic confidence thresholds and logic-based hierarchical fallback. Meanwhile, multi-layer attention maps from ViT serve as the basis for Position-Aware Attention Enhancement (PAAE), enabling position prompts and class-balanced attention supervision under noise and occlusion.

## 3 PROPOSED METHOD

This section presents **LRConfNet** (Logical Reasoning and Uncertainty Quantification-Driven Confidence Adjustment and Regularization Network), a hierarchical classification framework tailored for degraded images with uncertainty, logical inconsistency, and data imbalance. As shown in Figure 2, LRConfNet consists of four tightly coupled modules: (1) multi-level feature extraction via ViT, enhanced by Semantic-Guided Cross-Attention (SGCA) to inject label-tree priors and align visual–semantic representations across levels; (2) Logical Reasoning Regularization (LogReg), which enforces hierarchical consistency and inter-class feature alignment while deriving a complexity-aware confidence signal; (3) a logic-driven dynamic confidence adjustment mechanism that uses this signal to calibrate level-wise thresholds and trigger hierarchical fallback when fine-grained predictions are unreliable; and (4) Position-Aware Attention Enhancement (PAAE), which combines

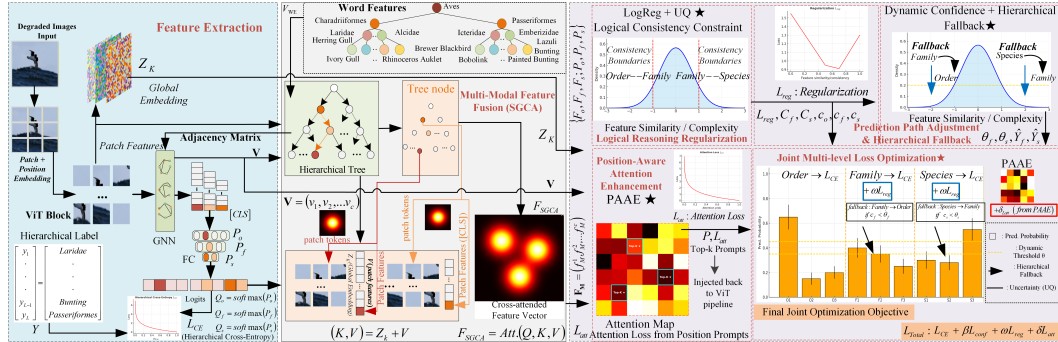

Figure 2: The Proposed LRConfNet: A Logical Reasoning and Uncertainty Quantification-Driven Confidence-Aware Hierarchical Classification Framework for Degraded Images. Visual representation of LRConfNet's architecture, illustrating its multi-modal feature extraction, logical reasoning-driven confidence adjustment, and adaptive hierarchical fallback for robust classification under degraded conditions.

class rebalancing and position prompts to keep attention on discriminative regions under noise, blur, low resolution, occlusion, and long-tail imbalance. Together, these modules form a unified hierarchical decision pipeline in which semantic alignment, uncertainty modeling, and decision control each address a distinct failure mode of degraded images while jointly improving the robustness and semantic consistency of hierarchical predictions.

*The code, datasets, and configuration files will be made publicly available.*

### 3.1 MULTI-MODAL FEATURE EXTRACTION WITH SEMANTIC GUIDANCE

Degraded images often weaken local visual cues and disrupt the alignment between appearance and the label hierarchy, so we first build a ViT-based multi-level feature extractor with SGCA to inject label-tree priors into visual features, obtain semantically aligned representations, and provide a reliable basis for downstream logical reasoning and confidence adjustment. Our model leverages ViT for multi-level feature extraction, effectively capturing global visual dependencies. Each input image $X \in \mathbb{R}^{H \times W \times C}$ is divided into $N$ non-overlapping patches, each encoded as $\mathbf{E}_{\text{patch}}$. These patches are projected into feature space and combined with positional encodings to form the initial Transformer input: $\mathbf{Z}_0 = \left[ \mathbf{E}_{\text{patch}}^1, \mathbf{E}_{\text{patch}}^2, \dots, \mathbf{E}_{\text{patch}}^N \right] + \mathbf{E}_{\text{pos}}$. These patch embeddings are then propagated through $K$ Transformer layers, each comprising Multi-Head Self-Attention (MSA) and Multi-Layer Perceptron (MLP) blocks: $\mathbf{Z}'_k = \text{MSA}\left( \text{LN}\left( \mathbf{Z}_{k-1} \right) \right) + \mathbf{Z}_{k-1}$. $\mathbf{Z}_k = \text{MLP}\left( \text{LN}\left( \mathbf{Z}'_k \right) \right) + \mathbf{Z}'_k, \quad k = 1, 2, \dots, K$, where $\text{LN}(\cdot)$ denotes layer normalization. The final global visual representation output is denoted as $\mathbf{Z}_K$. Here $k$ indexes Transformer layers (feature depth), to avoid confusion with $l$ reserved for taxonomy levels.

To further enhance this representation, motivated by the need to bridge visual features and semantic hierarchical relationships, especially under degraded conditions where semantic priors improve robustness and enforce hierarchical consistency. Specifically, SGCA leverages pre-trained GloVe (Global Vectors for Word Representation) embeddings (Pennington et al., 2014) to map class labels into a high-dimensional semantic space. These embeddings are defined as: $\mathbf{V} = (\mathbf{v}_1, \mathbf{v}_2, \dots, \mathbf{v}_c)$, where $c$ denotes the number of classes, and $\mathbf{v}_i$ represents the semantic embedding of the $i$-th class. Here, the semantic guidance is label-tree-based with textual priors from GloVe, complemented by hierarchical structure encodings. SGCA employs cross-attention to align semantic vectors $\mathbf{V}$ with visual features $\mathbf{Z}_K$, refining class-specific representations: $\mathbf{F}'_m = \text{MCA}\left( \text{LN}\left( \mathbf{F}_{m-1}, \mathbf{Z}_k \right) \right) + \mathbf{F}_{m-1}$. $\mathbf{F}_m = \text{MLP}\left( \text{LN}\left( \mathbf{F}'_m \right) \right) + \mathbf{F}'_m, \quad m = 1, 2, \dots, M$, where $\mathbf{F}_0 = \mathbf{V}$, and $M$ represents the number of SGCA layers. The final output $\mathbf{F}_M$ provides class-specific representations, which are further processed to generate attention-enhanced features $\mathbf{s} = \{s_1, s_2, \dots, s_c\}$ for HC.

## 3.2 LOGICAL REASONING REGULARIZATION AND UNCERTAINTY QUANTIFICATION

Most existing hierarchical classifiers handle each level independently, which can yield species-level predictions that contradict their family/order ancestors and offers no principled way to assess prediction reliability under degradation. LogReg addresses this by enforcing parent–child consistency in both probability and feature space while defining a feature-complexity–based uncertainty measure that drives the dynamic confidence adjustment in Sec. 3.3. HC inherently involves structured label spaces $Y = Y_1 \times \cdots \times Y_L$, where dependencies exist between levels $l \in \{1, \ldots, L\}$ (with $l$ indexing taxonomy levels, e.g., Order=1, Family=2, Species=3). Standard methods often neglect these dependencies, leading to inconsistent predictions. To this end, we introduce LogReg, a theoretically grounded framework inspired by (Tan et al., 2024), explicitly enforcing hierarchical structure while providing a principled basis for UQ.

HC inherently involves structured label spaces $Y = Y_1 \times \cdots \times Y_L$, where dependencies exist between levels $l \in \{1, \ldots, L\}$ (with $l$ indexing taxonomy levels, e.g., Order=1, Family=2, Species=3). Standard methods often neglect these dependencies, leading to inconsistent predictions. To this end, we introduce LogReg, a theoretically grounded framework inspired by (Tan et al., 2024), explicitly enforcing hierarchical structure while providing a principled basis for UQ.

**Hierarchical Consistency Constraints**. LogReg ensures predictions respect the hierarchical structure of categories. Specifically, we enforce that the confidence of a parent category should always be higher than or equal to its child category. For adjacent levels $l$ and $l-1$, this is defined as:

$$\mathcal{L}_c = \mathbb{E}_{(X,Y)\sim\mathcal{D}} \left[ \sum_{l=2}^{L} \lambda_{l,l-1} \cdot \text{ReLU} \left( P_l(y_l \mid X) - P_{l-1}(y_{l-1} \mid X) \right) \right] \quad (1)$$

where $P_l(y_l|X)$ is the predicted probability for category $y_l$ at level $l$, and $\lambda_{l,l-1}$ are hyperparameters controlling the strength of these constraints.

**Feature Consistency and Semantic Alignment**. Logical consistency should extend beyond probabilities to the feature space. We encourage feature alignment between adjacent levels using cosine similarity:

$$\mathcal{L}_s = \mathbb{E}_{X\sim\mathcal{D}} \Big[ \sum_{l=2}^{L} \bar{\lambda}_{l-1} \cdot \left( 1 - \frac{\bar{F}_l \cdot \bar{F}_{l-1}}{||\bar{F}_l|| \, ||\bar{F}_{l-1}||} \right) \Big] \quad (2)$$

where $\bar{F}_l$ denotes the batch-averaged feature representation at level $l$. This ensures that semantically related classes maintain consistent feature representations. Such geometric alignment provides the foundation for the logical constraint in Eq. 1, ensuring that probability consistency is supported by semantically coherent features.

**Uncertainty Quantification via Feature Complexity**. To robustly assess prediction reliability, we introduce a feature complexity-based UQ metric, defined as:

$$C(f_l') = \mathcal{H}_{\text{sample}}(f_l') + \lambda_{\text{var}} \cdot \sigma^2(f_l') \quad (3)$$

where $\mathcal{H}_{\text{sample}}$ denotes the entropy of the feature distribution, and $\sigma^2$ is the variance. Here $\mathcal{H}_{\text{sample}}(f_l')$ is computed by normalizing activations $f_l' = \text{softmax}(|f_l|)$, treating $f_l'$ as a probability vector, and then applying entropy $-\sum_{i=1}^{D_l} f_{l,i}' \log(\max(f_{l,i}', \epsilon))$. High entropy reflects uniform/uncertain features, while low entropy indicates sparse/discriminative activations. These complexity metrics are computed for both family and species levels:

$$C_f = \mathbb{E}_{f_{family}'}[C(f_{family}')], \quad C_s = \mathbb{E}_{f_{species}'}[C(f_{species}')] \quad (4)$$

This UQ measure allows LRConfNet to assess the reliability of its internal feature representations, enhancing its ability to make robust hierarchical decisions. For detailed derivations, including complete mathematical formulations of LogReg and UQ, please refer to Appendix A.5. As shown later in Table 6 and Table 12, LogReg mainly improves hierarchical precision while providing a stable uncertainty signal for the fallback mechanism.

## 3.3 LOGIC-DRIVEN DYNAMIC CONFIDENCE ADJUSTMENT AND HIERARCHICAL FALLBACK

Under severe degradation, fine-grained predictions can be over-confident yet unreliable, and fixed confidence thresholds cannot adapt to varying sample difficulty and degradation levels. We therefore use the feature-complexity score $C(\cdot)$ defined in Sec. 3.2 to adaptively set level-wise confidence

thresholds and trigger logic-driven hierarchical fallback, directly mitigating over-classification and improving hierarchical precision. Effective HC, especially under degraded image conditions, requires adaptive decision mechanisms that account for feature uncertainty. We introduce a **Logic-Driven Dynamic Confidence Adjustment** framework, calibrating decision thresholds based on feature complexity adaptively.

**Adaptive Confidence Thresholding**. To account for varying feature quality, we formulate a dynamic threshold $\theta_l^{(e)}$ for each hierarchical level $l \in \{\text{family}, \text{species}\}$:

$$\theta_l^{(e)} = \min\left(\max\left(\theta_{l,0} \cdot (1 + \gamma_l \cdot C_l^{(e)}), \theta_{\min}\right), \theta_{\max}\right) \quad (5)$$

where $\theta_{l,0}$ is the base threshold, $C_l^{(e)}$ denotes feature complexity as defined in Sec 3.2, $\gamma_l$ is a sensitivity coefficient for feature uncertainty. Here, the superscript $(e)$ indexes the training epoch, i.e., $C_l^{(e)}$ and $\theta_l^{(e)}$ are updated at epoch $e$; at inference time, we simply use the final thresholds $\theta_l = \theta_l^{(E)}$ and omit the superscript for brevity. This adaptive thresholding strategy directly relates the model's confidence threshold to the assessed feature quality, ensuring stricter requirements for fine-grained predictions when feature representations are uncertain.

**Logic-Driven Hierarchical Fallback Mechanism**. Based on these dynamically adjusted thresholds, we employ a hierarchical fallback strategy, defined as:

$$\hat{Y}_l = \begin{cases} Y_{\text{parent}(l)} & \text{if } p_l < \theta_l^{(e)} \\ Y_l & \text{if } p_l \geq \theta_l^{(e)} \end{cases} \quad (6)$$

This mechanism allows the model to revert to a higher-level category when fine-grained predictions are unreliable, maintaining semantic consistency, which is grounded in $C_l$. Note that Eq. 6 reassigns the model's *prediction output* based on confidence, rather than redefining the label domains. In practice, if a species prediction is below threshold, the model outputs its parent family prediction instead.

**Theoretical Justification and Practical Implications**. The adaptive confidence adjustment is derived from Bayesian decision-theoretic perspective (Wu & Cameron, 1990), where the optimal action minimizes expected risk:

$$a^* = \arg\min_a \mathbb{E}_{y_{\text{true}}|X}[\mathcal{L}(a, y_{\text{true}})] \quad (7)$$

This risk-aware formulation ensures that fine-grained classifications are accepted only when their confidence is justified by reliable feature representations. Crucially, our approach leverages $C_l$, derived from information theory principles (Gelman et al., 1995; Doshi-Velez & Kim, 2017; Tan et al., 2024), making it sensitive to image degradation. This dynamic confidence adjustment is further supported by LogReg in Sec 3.2, where semantic consistency across hierarchical levels is maintained.

### 3.4 POSITION-AWARE ATTENTION ENHANCEMENT FOR DEGRADED IMAGES

Under visual degradation and long-tail sparsity, local evidence for fine-grained recognition is often missing, causing the backbone to focus on noisy or background regions, especially for rare species. To address this, we introduce PAAE module that aggregates multi-layer attention, applies class-balanced re-weighting, and guides the final attention map toward the most informative patches, thereby improving fine-grained accuracy and robustness on degraded, imbalanced data. Accurately localizing discriminative features is particularly challenging in degraded images, as noise and artifacts obscure fine-grained details essential for HC. To this end, we introduce a PAAE module, integrating multi-layer attention analysis, class-balance weighting, and attention-guiding loss to focus on semantically important regions, inspired by (Dosovitskiy et al., 2020; Wang et al., 2023a; Abnar & Zuidema, 2020).

**Multi-Layer Attention Integration for Robust Salience Mapping**. To ensure consistent focus on critical regions, we aggregate attention weights across multiple transformer layers, leveraging the cumulative influence of all layers:

$$\mathcal{R} = \prod_{l=1}^{L} (\alpha \bar{\mathcal{A}}_l + (1 - \alpha) I_N) \quad (8)$$

where $\bar{\mathcal{A}}_l$ is the normalized attention matrix at layer $l$, $I_N$ is an identity matrix, and $\alpha$ is a blending factor. This approach is inspired by the attention rollout technique (Abnar & Zuidema, 2020), providing a stable focus map by capturing consistent attention across all layers. The final importance score vector $\mathbf{I} \in \mathbb{R}^{B \times N}$ is derived from $\mathcal{R}$ as $I_i = \frac{1}{N} \sum_{j=1}^{N} R_{ij}$, which reduces the impact of noise in individual layers and ensures consistent focus on discriminative regions, even in degraded conditions. These importance scores $\mathbf{I}$ are later combined with class weights (see Eq. 9) to form position prompts in Eq. 11), making explicit the connection between Eq. 8) and Eq. 11).

**Class-Balanced Attention Weighting**. To achieve a balanced objective, we re-weight the contribution of each sample based on its class frequency, effectively approximating the minimization of a balanced risk or maximizing a weighted likelihood (Huang et al., 2024; Wang et al., 2023a; Elkan, 2001). The weight $w_c$ for class $c$ is defined as:

$$w_c = \frac{(1/n_c)^\rho}{\sum_{j=1}^{C}(1/n_j)^\rho} \cdot C \tag{9}$$

Here, $n_c$ is the number of samples in class $c$, and $\rho = 0.5$ is a smoothing factor. This ensures that rare classes receive adequate attention, improving classification balance. For each sample $i$ with label $y_i$, its weight is $W_i = w_{y_i}$, which is then multiplied with the importance score $I_i$ in Eq. (11).

**Position Prompting and Attention Guidance Loss**. Leveraging the robust salience map $\mathbf{I}$ and class weights $W$, we generate Position Prompts $P \in \mathbb{R}^{B \times N}$ as a target spatial attention distribution:

$$P_{ij} = M_{K,ij} \cdot I_{ij} \cdot W_i \tag{10}$$

where $M_{K,ij}$ is a binary mask identifying the top-$K$ most salient patches, and $W_i$ denotes class balance weights derived from Eq. (9). To guide the model's focus (Cui et al., 2024), we introduce an Attention Loss:

$$\mathcal{L}_{\text{att}} = -\frac{1}{B} \sum_{i=1}^{B} \sum_{j=1}^{N} P'_{ij} \log \left( \frac{\exp(\mathcal{A}_{ij}^{(L)}/T)}{\sum_{k=1}^{N} \exp(\mathcal{A}_{ik}^{(L)}/T)} \right) \tag{11}$$

where $P'_{ij}$ is obtained by normalizing $P_{ij}$, $\mathcal{A}^{(L)}$ is the final layer's attention distribution, and $T$ is a temperature scaling factor. This loss aligns the model's attention with semantically informative regions, avoiding the domination of the classification process by degraded regions. PAAE reuses the attention maps already computed by the ViT encoder and only adds a training-time regularization loss; at inference LRConfNet performs the standard ViT forward pass without extra layers, so FLOPs and latency remain essentially unchanged compared to the backbone.

### 3.5 JOINT OPTIMIZATION OBJECTIVE

Effective training of LRConfNet requires a comprehensive optimization objective that not only ensures classification accuracy but also maintains hierarchical consistency, calibrates confidence estimates, and focuses on discriminative regions. To achieve this, we propose a joint optimization objective, which integrates multiple loss components, each addressing critical aspects of robust HC:

**Hierarchical Cross-Entropy Loss**. This forms the primary supervised signal for classification accuracy across the hierarchy $l \in \{\text{level}, \text{order}, \text{family}, \text{species}\}$. Crucially, it operates on the fallback-adjusted targets $\hat{y}_l$ derived from the logic-driven dynamic confidence mechanism Eq. 6 (with thresholds from Eq. 5), ensuring that the loss computation reflects the adaptively chosen prediction path. Let $P_l = \text{softmax}(\text{logits}_l)$ be the predicted probability distribution at level $l$, $N_l$ be the number of samples active at level $l$ in the batch (considering hierarchy), $C_l$ be the number of classes at level $l$, and $y_{ij}^{(l)}$ be the one-hot ground-truth (or adjusted target $\hat{y}_l$) for sample $i$ class $j$. The formulation is:

$$\mathcal{L}_{CE}(\Theta; X, Y) = \sum_{l \in \{\text{level, order, family, species}\}} \lambda_l \cdot \left( -\frac{1}{N_l} \sum_{i=1}^{N} \sum_{j=1}^{C_l} y_{lj}^{(l)} \log P_l(j|X_i; \Theta) \right) \tag{12}$$

where $\lambda_l$ are hyperparameters balancing the influence of each hierarchical level. This component directly optimizes predictive accuracy based on the uncertainty-aware classification path.

**Confidence Regularization Loss**. To explicitly encourage the model to produce reliable confidence estimates $(p_f, p_s)$ and penalize overconfident errors or underconfident correct predictions, especially

at finer levels susceptible to degradation impact, we incorporate a confidence loss term. Drawing from calibration and uncertainty literature (Guo et al., 2017), we adopt a squared error against ideal confidence (1.0):

$$\mathcal{L}_{conf}(\Theta; X, Y) = \mathbb{E}_{(X,Y)} \left[ \sum_{l \in \{\text{family, species}\}} (1 - P_l(\hat{y}_l | X; \Theta))^2 \right] \tag{13}$$

where $P_l(\hat{y}_l | X; \Theta)$ is the predicted confidence for the (potentially fallback-adjusted) target class $\hat{y}_l$. This loss incentivizes the model to align its output probabilities with true correctness likelihoods. A dynamically decaying weight $w(e) = \max(0.1, 1.0 - e/E)$ is applied during training.

Combining these terms, the final Joint Optimization Objective is:

$$\min_{\Theta} \mathbb{E}_{(X,Y) \sim \mathcal{D}} \left[ \mathcal{L}_{CE}(\Theta; X, Y) + \beta w(e) \mathcal{L}_{conf}(\Theta; X, Y) + \omega \mathcal{L}_{reg}(\Theta; X, Y) + \delta \mathcal{L}_{att}(\Theta; X, Y) \right] \tag{14}$$

where $\beta, \omega, \delta$ are hyperparameters balancing the trade-offs between the primary classification task and the auxiliary components for uncertainty, logical consistency, and visual localization. Here, $\mathcal{L}_{reg} = \mathcal{L}_c + \mathcal{L}_s$ serves as a structural prior from the hierarchical domain knowledge.

## 4 EXPERIMENTS AND DISCUSSION

### 4.1 EXPERIMENTAL DATA AND SETUP

To facilitate rigorous evaluation, we introduce two new hierarchical benchmarks, HRSC-Deg and CUB-Deg, derived by post-processing the predictions of a pre-trained hierarchical classifier (Chang et al., 2021) on HRSC (Liu et al., 2017) and CUB-200-2011 (Wah et al., 2011). This construction enforces hierarchical consistency and significantly reduces annotation overhead. The HRSC dataset contains remote sensing images of ships with resolutions ranging from $300 \times 300$ to $1500 \times 900$, annotated with a three-level label hierarchy. For increased complexity, we utilize a two-level taxonomy consisting of 3 coarse-grained and 21 fine-grained ship categories. The CUB dataset comprises 200 bird species, hierarchically organized into 13 orders, 38 families, and 200 species based on Wikipedia's biological taxonomy. To ensure hierarchical consistency and reduce manual annotation costs, we apply a pre-trained automated hierarchical classifier (Chang et al., 2021) to relabel the original HRSC and CUB annotations. We then filter inconsistent labels to construct the HRSC-Deg and CUB-Deg datasets. Additionally, we introduce controlled data degradation using the augmentation function $G(t, \sigma, \eta, \lambda, \delta)$ (Wang et al., 2023b), incorporating noise injection, blurring, cropping, and other transformations (details provided in Table 3 of Appendix A.1). Samples violating hierarchical constraints are excluded. All HRSC-Deg images are resized to $224 \times 224$, and all CUB-Deg images to $448 \times 448$. We use a ViT-Base/16 backbone initialized with ImageNet-1K pre-trained weights and fine-tune all layers jointly. Training configurations and evaluation metrics are provided in Appendices A.1 and A.2.

LRConfNet keeps the ViT backbone architecture unchanged, and all additional components operate on low-dimensional feature statistics or logits. In our implementation, the parameter count increases from 86.4M to 123.9M and the per-image inference time (batch size 1) from $34.28 \pm 2.12$ ms to $35.70 \pm 7.76$ ms (only $\approx 4.1\%$ slower, still about 28 FPS), confirming that the computational cost remains dominated by the ViT encoder.

### 4.2 EXPERIMENTAL RESULTS AND ANALYSIS

Table 1 compares our LRConfNet with state-of-the-art methods (Cerri et al., 2016; Wehrmann et al., 2018; Chang et al., 2021; Dosovitskiy et al., 2020; Chen et al., 2022; Jain et al., 2023; Wang et al., 2023c;b; Shan et al., 2024; Ke et al., 2022; Goren et al., 2024) on two cross-domain degraded image datasets. These datasets span both remote sensing and natural image domains, incorporating image quality degradation and complex HC structures (two-level for HRSC-Deg and three-level for CUB-Deg), significantly increasing the classification challenge.

On HRSC-Deg, LRConfNet achieves 95.07% in $P_H$ and 94.04% in $R_H$, surpassing the closest competitor (HSC) by 3.03% and 3.61%. Notably, it attains 91.76% in *Fine*-grained classification,

Table 1: Comparison with State-of-the-Art Methods

| Methods | HRSC-Deg | | | | Class_acc | | CUB-Deg | | | | Class_acc | | |
|---|---|---|---|---|---|---|---|---|---|---|---|---|---|
| | $ISDL$ | $P_H$ | $R_H$ | $Lvl_{acc}$ | Coarse | Fine | $ISDL$ | $P_H$ | $R_H$ | $Lvl_{acc}$ | Order | Family | Species |
| HMC-LMLP | 68.57 | 86.87 | 88.79 | 73.08 | 98.02 | 83.28 | 40.04 | 73.09 | 71.57 | 70.66 | 95.67 | 70.72 | 25.08 |
| HMCN | 70.19 | 88.57 | 89.62 | 74.40 | 98.81 | 84.32 | 36.72 | 68.24 | 68.49 | 69.38 | 95.62 | 62.45 | 21.25 |
| CH-PMG | 72.08 | 90.49 | 89.12 | 74.84 | 98.21 | 89.20 | 54.69 | 83.68 | 81.95 | 71.92 | 97.63 | 85.46 | 69.84 |
| ViT-B | 66.03 | 89.01 | 88.62 | 72.21 | 97.88 | 84.98 | 47.43 | 80.39 | 80.86 | 71.06 | 90.83 | 78.55 | 61.58 |
| HRN | 72.46 | 90.88 | 89.45 | 75.82 | 99.10 | 88.85 | 55.23 | 85.42 | 84.41 | 72.99 | 98.15 | 86.41 | 70.11 |
| HiE | 69.27 | 90.93 | 88.22 | 74.00 | 98.51 | 88.40 | 61.98 | 85.79 | 85.88 | 73.01 | 98.67 | 87.34 | 74.89 |
| TransHP | 71.90 | 91.37 | 89.27 | 74.12 | 98.81 | 88.58 | 62.74 | 87.66 | 87.64 | 73.12 | 98.40 | 90.13 | 76.78 |
| SGHPN | 72.73 | 91.07 | 90.51 | 75.89 | 98.48 | 88.87 | 65.62 | 90.71 | 92.84 | 72.99 | 98.36 | 90.11 | 79.54 |
| GvT | 68.92 | 89.45 | 89.12 | 74.00 | 98.18 | 85.69 | 62.79 | 88.87 | 90.98 | 71.65 | 98.43 | 82.09 | 79.54 |
| CAST | 72.01 | 91.76 | 90.04 | 74.23 | 99.10 | 88.75 | 62.91 | 89.22 | 89.78 | 73.09 | 98.51 | 90.88 | 78.93 |
| HSC | 72.23 | 92.04 | 90.43 | 74.34 | 99.40 | 88.93 | 64.01 | 90.04 | 91.01 | 73.16 | 98.69 | 91.79 | 81.17 |
| LRConfNet(Ours) | **74.10** | **95.07** | **94.04** | **77.81** | **99.80** | **91.76** | **66.31** | **91.78** | **93.60** | **74.76** | **99.02** | **96.06** | **85.03** |

highlighting the effectiveness of our UQ-driven dynamic confidence adjustment and logical reasoning regularization. On the more challenging CUB-Deg dataset, LRConfNet further excels, achieving 85.03% at the *Species* level—3.86% higher than the best baseline—and 66.31% in $ISDL$, demonstrating superior hierarchical consistency. This performance is driven by our multi-level loss optimization and attention-guided enhancement mechanism, which enhance feature localization.

Overall, these results underscore LRConfNet's superior capability in robust hierarchical classification, effectively navigating complex hierarchical structures under severe image degradation. The model's UQ with semantic priors and attention-guided enhancement further strengthen its focus on discriminative regions, achieving superior fine-grained classification performance.

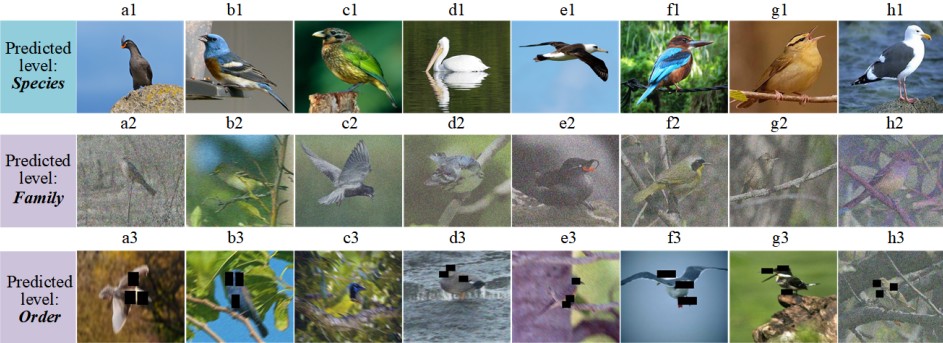

Figure 3: Adaptive Hierarchical Prediction with Semantically Guided Confidence Adjustment and Hierarchical Fallback. From the first row (a1-h1) to the third row (a3-h3), the image quality progressively deteriorates. Our hierarchical predictor effectively selects fine-grained levels (species) for high-quality images, while adaptively reverting to coarse-grained levels (family or order) for low-quality images. This illustrates the model's adaptive confidence adjustment and robust hierarchical fallback mechanism.

Figure 3 shows that LRConfNet adaptively selects the prediction level by image quality, yielding robust hierarchical classification. Using SGCA and dynamic confidence adjustment, the model chooses the optimal level—species, family, or order. The first row (a1–h1) presents accurate species-level predictions for high-quality images; the second (a2–h2) shows family-level predictions for moderately degraded images where fine-grained cues are unreliable; the third (a3–h3) shows order-level predictions for severely degraded images, where hierarchical fallback avoids over-classification. Taken together, these examples constitute qualitative hierarchical decision paths under increasing degradation, showing how our confidence-aware fallback mitigates error propagation and preserves semantic consistency, while remaining failure cases—e.g., confusion between extremely similar ship types or bird species, or overly conservative fallback to a nearby parent class under extreme blur and occlusion—also delineate the current limits of LRConfNet.

Table 2: Ablation Study on LogReg and PAAE Mechanisms on HRSC-Deg and CUB-Deg.

| Methods | | HRSC-Deg | | | | | | CUB-Deg | | | | | | |
|---|---|---|---|---|---|---|---|---|---|---|---|---|---|---|
| LogReg | PAAE | $ISDL$ | $P_H$ | $R_H$ | $Lvl_{acc}$ | *Class_acc* | | $ISDL$ | $P_H$ | $R_H$ | $Lvl_{acc}$ | *Class_acc* | | |
| | | | | | | *Coarse* | *Fine* | | | | | *Order* | *Family* | *Species* |
| ✗ | ✗ | 71.94 | 92.86 | 90.68 | 75.11 | 99.10 | 89.40 | 65.67 | 90.41 | 91.97 | 73.19 | 98.77 | 94.19 | 83.26 |
| ✗ | ✓ | 73.39 | 94.17 | 92.60 | 76.15 | 99.41 | 89.94 | 65.71 | 90.44 | 92.98 | 73.34 | 98.96 | 94.96 | 84.82 |
| ✓ | ✗ | 73.14 | 93.35 | 91.44 | 76.68 | 99.12 | 89.49 | 65.95 | 90.79 | 92.91 | 73.60 | 98.93 | 94.96 | 84.71 |
| ✓ | ✓ | **74.10** | **95.07** | **94.04** | **77.81** | **99.80** | **91.76** | **66.31** | **91.78** | **93.60** | **74.76** | **99.02** | **96.06** | **85.03** |

### 4.3 Ablation Study

We conducted ablation experiments to evaluate the contributions of LogReg and PAAE to HC performance. Table 2 presents the results, demonstrating how each component mitigates the challenges of degraded image classification.

**1) Effectiveness of LogReg and PAAE.** The baseline without LogReg and PAAE attains 71.94% $ISDL$ and 89.40% fine-grained accuracy on HRSC-Deg, and 83.26% species accuracy on CUB-Deg. Adding PAAE consistently increases fine-grained (HRSC-Deg) and species-level (CUB-Deg) accuracy, indicating improved focus on critical regions. LogReg further raises level accuracy on both datasets, validating the benefit of enforcing hierarchical consistency. Additional attention-rollout visualizations in Figure 6 further show that PAAE suppresses background noise and steers the model toward discriminative bird regions under severe degradation, providing the fine-grained evidence required for reliable hierarchical decisions.

**2) Combined Impact of LogReg and PAAE.** The combined model (LogReg + PAAE) delivers the highest performance, achieving 74.10% $ISDL$, 95.07% $P_H$, and 94.04% $R_H$ on HRSC-Deg, and 85.03% species-level accuracy on CUB-Deg. The largest gains appear at the fine-grained level, indicating that LogReg enforces semantic consistency while PAAE sharpens localization, resulting in a robust HC model under challenging conditions.

**3) Hyperparameter sensitivity.** Beyond the above ablations, we further conduct a systematic sensitivity study of key hyperparameters in Appendix A.9.3. By varying the LogReg weights, the dynamic threshold coefficients ($\theta_{l,0}$, $\gamma_l$, $\theta_{\min}$, $\theta_{\max}$), and the PAAE settings ($\rho$, $K$, $T$) within reasonably wide ranges, we observe smooth changes in ISDL, hierarchical precision/recall, and fine-grained accuracy (see Appendix A.9.3, Tables 9– 12 and Figures 7– 10), without abrupt performance drops. These results indicate that LRConfNet is reasonably robust to hyperparameter choices around our default configuration.

## 5 Conclusion

We propose LRConfNet, a UQ- and logical-reasoning–driven framework with dynamic confidence adjustment for robust hierarchical classification of remote-sensing and natural images. By coupling uncertainty estimation with a logic-guided fallback, LRConfNet yields consistent, adaptive decisions under severe degradation. A ViT backbone with Semantic-Guided Cross-Attention fuses visual and semantic cues, while LogReg enforces taxonomy consistency. The confidence-aware fallback limits error propagation under class imbalance and hierarchical sparsity. A joint objective—combining position-aware attention, attention–gradient fusion, and level-specific weighting—improves feature selection across coarse and fine levels. On cross-domain degraded benchmarks, LRConfNet surpasses state of the art, delivering higher hierarchical consistency and fine-grained accuracy under strong degradations. Future work will assess LRConfNet on real-world degraded imagery and under stronger domain shifts, extend it to other domains such as medical or document HC, and combine it with open-set or out-of-hierarchy detection and richer uncertainty modeling, leveraging its reliance only on a tree-structured label space rather than domain-specific priors.

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

# A APPENDIX

## A.1 EXPERIMENTAL DATA AND EXPERIMENTAL SETUP

The dataset used in this study is derived from HRSC (Liu et al., 2017), with further pre-processing and new data generation to meet the experimental requirements. The dataset comprises remote sensing ship images of varying scales and resolutions ($300 \times 300$ to $1500 \times 900$), hierarchically annotated into three levels: coarse-grained categories (e.g., *military*, *commercial*, *other*), mid-level types (e.g., aircraft carrier, merchant vessel), and fine-grained models (e.g., *Nimitz-class*, *Tarawa-class*). To reduce manual labeling costs, we employed an automatic hierarchical classification (HC) model based on Chang et al. (Chang et al., 2021). The model was pre-trained on the original HRSC annotations and used to automatically label newly generated samples. The predicted hierarchical labels were then reviewed and refined to ensure consistency and integrity across levels.

To further enhance the generalization and robustness of our model, we introduced artificial degradation via a dataset augmentation function $G(t, \sigma, \eta, \lambda, \delta)$ as defined in (Wang et al., 2023b). The augmentations include noise injection, image blurring, and random cropping, where $\sigma$, $\eta$, $\lambda$, and $\delta$ control the strength of each transformation. Table 3 summarizes the parameter settings for these augmentations. The augmented samples were filtered using the HC model to remove logically inconsistent annotations. For instance, if a sample is labeled as "Nimitz-class aircraft carrier" in fine-grained prediction but "commercial ship" in coarse-grained prediction, it is discarded. Only samples that satisfy hierarchical constraints—e.g., encoded as $s = [1, 1]$ or $[1, 0]$, where 1 denotes correct classification—are retained. To increase complexity, we use two levels—3 coarse-grained and 21 fine-grained categories. Figure 4 presents representative examples of both clean and degraded images, with corresponding distortion parameters summarized in Table 4.

In parallel, we construct the CUB-Deg dataset based on CUB-200-2011 (Wah et al., 2011), which contains 11,788 bird images across 200 species. We restructured the taxonomy into a three-level hierarchy (order, family, species) and generated hierarchical labels accordingly. To simulate real-world degradation, we applied the same augmentation strategy $G(t, \sigma, \eta, \lambda, \delta)$ used for HRSC-Deg. The degradation parameters were randomly sampled to introduce variability in resolution, occlusion, and noise across the dataset. Similar to HRSC-Deg, samples that violate hierarchical consistency were filtered out. The final CUB-Deg dataset consists of 5994 training and 5794 testing samples, comprising 1960/1054/2980 instances (order/family/species) for training, and 1895/1052/2847 for testing, with the same multi-level annotation structure. Figure 4 and 5show examples of original and degraded imagess from both domains.

These two datasets from distinct domains—remote sensing and natural fine-grained recognition—enable comprehensive evaluation of our model's robustness and adaptability under complex, uncertain, and hierarchically structured conditions.

Figure 4: Examples of original and degraded imagess from two domains: HRSC-Deg (a–i) for ships. Distortions include noise, occlusion, and resolution degradation, simulating real-world scenarios and highlighting the challenges of fine-grained and HC under adverse conditions.

Table 3: Parameterized Degradation Simulation Methods for Robustness-Aware Data Augmentation

| Method | Description | Parameters / Range |
|---|---|---|
| White noise | Adds Gaussian white noise to the R, G, B channels of the image with the same standard deviation $\sigma$. | Standard deviation $\sigma$: 0.1 to 0.2 |
| Motion blur | Applies motion blur to the R, G, B channels of the image using a directional kernel. The direction is randomly selected from $0°$ to $180°$ | Kernel size $\eta$: 5 to 14; Direction: $0°$ to $180°$ |
| Downsampling | Simulates low-resolution images by downsampling the image. The downsampling rate $\lambda$ is defined as the ratio of the output image area to the original image area. | Downsampling rate $\lambda$: 0.4 to 0.7 |
| Cutout | Simulates occlusion by randomly selecting one of five positions (center, left, right, top, bottom) within important regions of the ship image and applying zero-masking. | Occlusion position: randomly selected |

Model training is performed using SGD with a momentum of 0.9, weight decay of $5 \times 10^{-4}$, and learning rate decay by a factor of 10 at epochs 25 and 40. For HRSC-Deg, we use a learning rate of 0.002 and batch size of 32; for CUB-Deg, we set the learning rate to 0.0001 and batch size to 16. Each model is trained for 100 epochs using PyTorch on NVIDIA V100 GPUs. We employ standard augmentations such as flipping and cropping and integrate hierarchical constraints during training to improve robustness and suppress label noise.

## A.2 EVALUATION METRICS

In HC, traditional metrics such as accuracy, recall, and $F_1$ score may fail to fully capture the hierarchical and semantic aspects of predictions. To provide a more comprehensive evaluation, we

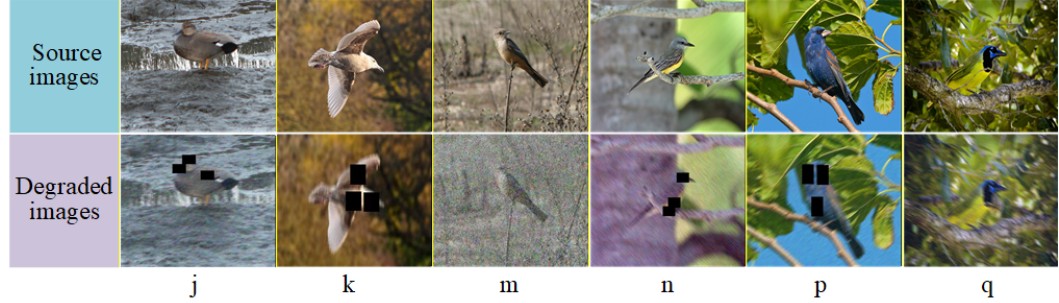

Figure 5: Examples of original and degraded imagess from CUB-Deg (j–q) for bird species. Distortions include noise, occlusion, and resolution degradation, simulating real-world scenarios and highlighting the challenges of fine-grained and HC under adverse conditions.

Table 4: Examples of Distortion Parameters for Different Image Data.

| | HRSC-Deg | | | | | CUB-Deg | | | |
| --- | --- | --- | --- | --- | --- | --- | --- | --- | --- |
| Degraded img. No. | $\sigma$ | $\delta$ | $\eta$ | $\lambda$ | Degraded img. No. | $\sigma$ | $\delta$ | $\eta$ | $\lambda$ |
| a | - | 0.0294 | - | - | j | 0.1955 | 0.0400 | 10 | 0.5710 |
| b | 0.1612 | - | 7 | - | k | - | 0.0385 | 13 | 0.4153 |
| c | 0.1958 | 0.0333 | - | 0.7583 | m | 0.1829 | - | - | 0.4142 |
| d | - | - | 10 | 0.7457 | n | 0.1815 | 0.0323 | 11 | 0.6511 |
| e | 0.1347 | 0.0400 | - | - | p | 0.1312 | 0.0400 | 13 | - |
| f | - | 0.0400 | - | 0.6255 | q | 0.1291 | - | 14 | - |
| g | - | - | 9 | - | - | - | - | - | - |
| h | - | - | - | 0.7470 | - | - | - | - | - |
| i | 0.1155 | - | 12 | 0.7417 | - | - | - | - | - |

Note: The letters "a"–"i", "j"–"k" and "m"–"q" represent randomly selected samples from the newly generated dataset.

adopt six metrics: Inverse Symmetric Difference Loss ($ISDL$), Hierarchical Precision ($P_H$), Hierarchical Recall ($R_H$), Hierarchical Accuracy, Coarse-grained Accuracy, and Fine-grained Accuracy. $ISDL$ measures the structural consistency between prediction and ground truth and is defined as $ISDL = \frac{1}{1+|(S \setminus \hat{S}) \cup (\hat{S} \setminus S)|}$, where $S$ and $\hat{S}$ denote the sets of ground-truth and predicted labels, respectively. A higher $ISDL$ indicates better alignment between predicted and true label sets. The hierarchical precision and recall are computed as $P_H = \frac{|S \cap \hat{S}|}{|\hat{S}|}$ and $R_H = \frac{|S \cap \hat{S}|}{|S|}$, where $|S \cap \hat{S}|$ is the number of correctly predicted labels across all hierarchical levels. To capture level-specific performance, we additionally report HRSC-Deg using $Acc_{coarse}$ and $Acc_{fine}$, and CUB-Deg using $Acc_{order}$, $Acc_{family}$, and $Acc_{species}$. The overall hierarchical accuracy $Lvl_{acc}$ is defined as the mean accuracy across all levels. Together, these metrics assess the model's consistency, robustness, and its ability to make semantically meaningful predictions at multiple levels of granularity.

### A.3 MORE DISCUSSION ON PROPOSED METHOD

To provide a clear understanding of LRConfNet's design and functionality, we present Algorithm 1, detailing a comprehensive framework that enhances HC under degraded image conditions. This framework integrates logical reasoning regularization, UQ, and dynamic confidence adjustment, forming a robust approach for multi-level classification. A key innovation is the dynamic confidence weight decay (lines 3-4), which adaptively balances the influence of confidence penalties during training, ensuring consistent confidence calibration. The framework quantifies feature complexity using entropy and variance metrics (lines 16-18), capturing both aleatoric and epistemic uncertainty. The hierarchical fallback mechanism (line 22) is particularly impactful, allowing low-confidence predictions at the species level to revert to higher-level categories, such as family or order, preserv-

ing semantic consistency. Additionally, the attention enhancement module (lines 25-28) leverages position prompts to guide model focus, ensuring attention is concentrated on semantically relevant regions. This multi-loss optimization strategy (line 29) effectively balances classification accuracy, confidence calibration, logical consistency, and attention enhancement, making LRConfNet highly effective in fine-grained recognition tasks under challenging image conditions.

---

**Algorithm 1** LRConfNet: Logical Reasoning-Driven Confidence Adjustment

---

**Input**: Training dataset $D$, initial thresholds $\theta_s, \theta_f$,
    complexity coefficients $\gamma_f, \gamma_s$, loss weights $\beta, \omega, \delta$, temperature $T = 1.0$

**Output**: Optimized model parameters $\Theta^*$ for HC with logical reasoning

  1: Initialize ViT, SGCA module and classification heads parameters in $\Theta \sim \mathcal{N}(0, 0.01)$
  2: $\theta_s \leftarrow \theta_{s,0}, \theta_f \leftarrow \theta_{f,0}$                                                ▷ Species/Family confidence thresholds
  3: **for** each epoch $e$ from 1 to $E$ **do**
  4:     $w(e) \leftarrow \max(0.1, 1.0 - \frac{e}{E})$                                 ▷ Dynamic confidence weight decay
  5:     **for** each batch $(X, Y_l, Y_o, Y_f, Y_s) \in D$ **do**
  6:        $Z_K, V \leftarrow \text{ViT}(X)$                            ▷ Global embedding & patch features
  7:        $F_1, F_2, F_L \leftarrow \text{SGCA}(Z_K, V, \text{trees})$            ▷ Cross-attention modal fusion
  8:        $[P_l, P_o, P_f, P_s, c_s, c_f, c_o] \leftarrow \Theta(F_1, F_2, F_L)$    ▷ Multi-level predictions & confidences
  9:        $Q_o \leftarrow \text{softmax}(P_o), Q_f \leftarrow \text{softmax}(P_f), Q_s \leftarrow \text{softmax}(P_s)$
10:        $c_o \leftarrow Q_o[\text{batch\_idx}, Y_o], c_f \leftarrow Q_f[\text{batch\_idx}, Y_f], c_s \leftarrow Q_s[\text{batch\_idx}, Y_s]$
11:        $\mathcal{L}_c \leftarrow \text{ReLU}(c_f - c_o).\text{mean}() + 1.5 \cdot \text{ReLU}(c_s - c_f).\text{mean}()$      ▷ Hierarchical consistency
12:        $F_L \leftarrow \text{mean}(F_L, \dim = 1)$
13:        $S_{fs} \leftarrow \text{cosine\_similarity}(F_{L-1}, F_L)$
14:        $\mathcal{L}_s \leftarrow 0.5 \cdot (1 - S_{of}).\text{mean}() + 0.8 \cdot (1 - S_{fs}).\text{mean}()$          ▷ Feature similarity
15:        $F_L' \leftarrow \text{normalize}(F_L, p = 2)$
16:        $H_s \leftarrow -\sum(F_L' \cdot \log(\max(F_L', 1e - 10)), \dim = 1)$         ▷ Feature entropy
17:        $\sigma_s^2 \leftarrow \text{var}(F_L', \dim = 1)$                           ▷ Feature variance
18:        $C_s \leftarrow \text{mean}(H_s + 0.5 \cdot \sigma_s^2)$               ▷ Complexity scores
19:        $\mathcal{L}_{reg} \leftarrow \mathcal{L}_c + \mathcal{L}_s$            ▷ Combined logical reasoning regularization loss
20:        $F_s \leftarrow 1.0 + \gamma_s \cdot C_s$               ▷ Complexity-based factors
21:        $\theta_s \leftarrow \min(\max(\theta_s \cdot F_s, 0.5), 0.85)$         ▷ Adaptive thresholds
22:        $\hat{Y}_s \leftarrow (Y_l = 3 \,\&\&\, c_s < \theta_s)?Y_f : Y_s$
23:        $\hat{Y}_f \leftarrow (Y_l = 2 \,\&\&\, c_f < \theta_f)?Y_o : Y_f$         ▷ Hierarchical backtracking
24:        $\mathcal{L}_{CE} \leftarrow \text{CE}(P_l, Y_l) + \text{CE}(P_o, Y_o) + \text{CE}(P_f, \hat{Y}_f) + \text{CE}(P_s, \hat{Y}_s)$
25:        $\mathcal{L}_{conf} \leftarrow \text{mean}((1 - c_s)^2 + (1 - c_f)^2)$         ▷ Confidence penalty
26:        $I \leftarrow \text{Attention Importance}(\Theta.\text{attention\_maps})$       ▷ Patch importance scores
27:        $W \leftarrow \text{Class Balance Weights}(Y_s, \text{num\_classes} = 200)$    ▷ Address class imbalance
28:        $P \leftarrow \text{TopK}(I, k) \cdot I \cdot W[Y_s]$         ▷ Position prompts generation
29:        $\mathcal{L}_{att} \leftarrow -\sum_i P_i \cdot \log\left(\text{softmax}(\Theta.\text{attention\_maps}[-1][:, 0, 1 :]/T)_i\right)$    ▷ Attention loss
30:        $\mathcal{L}_{total} \leftarrow \mathcal{L}_{CE} + \beta \cdot w(e) \cdot \mathcal{L}_{conf} + \omega \cdot \mathcal{L}_{reg} + \delta \cdot \mathcal{L}_{att}$
31:        $\nabla\Theta \leftarrow \frac{\partial \mathcal{L}_{total}}{\partial \Theta}, \Theta \leftarrow \Theta - \eta \cdot \nabla\Theta$
32:     **end for**
33:     $A_l, A_o, A_f, A_s, ISDL, P_H, R_H \leftarrow \text{Evaluate Hierarchical}(\text{model})$
34: **end for**
35: **return** $\Theta^*$

---

We present Algorithm 2, which details the Logical Reasoning with UQ module, an enhancement of the LogReg framework that incorporates feature complexity analysis, enabling the model to dynamically assess uncertainty and maintain semantic consistency across hierarchical levels. The algorithm begins by extracting confidence probabilities for target categories across hierarchical levels (lines 2-3) and employs L2 normalization to standardize feature representations (lines 4-5). This enables the computation of complexity scores based on entropy and variance (lines 6-7), effectively capturing both aleatoric and epistemic uncertainty. A core innovation of this module is the dual-constraint regularization strategy: hierarchical consistency is enforced through weighted ReLU operations (lines 8-10), while feature similarity is maintained across adjacent taxonomic levels (lines 11-13). This dual regularization ensures that the model not only maintains logical consistency in its predictions but also learns semantically coherent feature representations, enhancing generalization under diverse image conditions.

To ensure reliable HC under uncertain conditions, Algorithm 3 implements a logic-driven hierarchical fallback optimization mechanism, providing dynamic confidence adjustment and adaptive

---

**Algorithm 2** Logical Reasoning with Uncertainty Quantification

---

**Input**: Order features $\mathbf{F}_1 \in \mathbb{R}^{B \times D_1}$, Family features $\mathbf{F}_2 \in \mathbb{R}^{B \times D_2}$, Species features $\mathbf{F}_L \in \mathbb{R}^{B \times D_L}$,
   Order logits $\mathbf{P}_o \in \mathbb{R}^{B \times C_1}$, Family logits $\mathbf{P}_f \in \mathbb{R}^{B \times C_2}$, Species logits $\mathbf{P}_s \in \mathbb{R}^{B \times C_L}$,
   Order labels $\mathbf{Y}_o$, Family labels $\mathbf{Y}_f$, Species labels $\mathbf{Y}_s$

**Output**: LogReg Loss $\mathcal{L}_{reg}$, Feature complexity scores $C_f$ & $C_s$, Target confidences $c_o, c_f, c_s$

---

1: **function** LOGICAL REASONING WITH UQ($\mathbf{F}_1, \mathbf{F}_2, \mathbf{F}_L, \mathbf{P}_o, \mathbf{P}_f, \mathbf{P}_s, \mathbf{Y}_o, \mathbf{Y}_f, \mathbf{Y}_s$)
2:    $\mathbf{Q} \leftarrow [\text{softmax}(\mathbf{P}_o), \text{softmax}(\mathbf{P}_f), \text{softmax}(\mathbf{P}_s)]$         ▷ Softmax normalization
3:    $[c_o, c_f, c_s] \leftarrow [\mathbf{Q}[0][\text{range}(B), \mathbf{Y}_o], \mathbf{Q}[1][\text{range}(B), \mathbf{Y}_f], \mathbf{Q}[2][\text{range}(B), \mathbf{Y}_s]]$
4:    $\mathbf{F}' \leftarrow [\text{softmax}(|\mathbf{F}_2|), \text{softmax}(|\mathbf{F}_L|)]$
5:    $\mathbf{H} \leftarrow [-\sum \mathbf{F}'[0] \log(\max(\mathbf{F}'[0], 1e^{-10})),$
       $-\sum \mathbf{F}'[1] \log(\max(\mathbf{F}'[1], 1e^{-10}))]$         ▷ Entropy of family and species features
6:    $\sigma^2 \leftarrow [\text{var}(\mathbf{F}'[0]), \text{var}(\mathbf{F}'[1])]$         ▷ Feature variance
7:    $[C_f, C_s] \leftarrow [\text{mean}(H[0] + 0.5\sigma^2[0]), \text{mean}(H[1] + 0.5\sigma^2[1])]$     ▷ Feature complexity scores
8:    $\mathcal{L}_{of} \leftarrow \text{mean}(\text{ReLU}(c_f - c_o))$         ▷ Order-Family consistency
9:    $\mathcal{L}_{fs} \leftarrow \text{mean}(\text{ReLU}(c_s - c_f))$         ▷ Family-Species consistency
10:    $\mathcal{L}_c \leftarrow \mathcal{L}_{of} + 1.5 \times \mathcal{L}_{fs}$         ▷ Weighted consistency loss
11:    $\overline{\mathbf{F}} \leftarrow [\text{mean}(\mathbf{F}_1), \text{mean}(\mathbf{F}_2), \text{mean}(\mathbf{F}_L)]$         ▷ Feature means
12:    $\mathbf{S} \leftarrow [\frac{\overline{\mathbf{F}}[0] \cdot \overline{\mathbf{F}}[1]}{||\overline{\mathbf{F}}[0]|| \cdot ||\overline{\mathbf{F}}[1]||}, \frac{\overline{\mathbf{F}}[1] \cdot \overline{\mathbf{F}}[2]}{||\overline{\mathbf{F}}[1]|| \cdot ||\overline{\mathbf{F}}[2]||}]$
13:    $\mathcal{L}_s \leftarrow 0.5 \times (1 - S[0]) + 0.8 \times (1 - S[1])$         ▷ Weighted similarity loss
14:    $\mathcal{L}_{reg} \leftarrow \mathcal{L}_c + \mathcal{L}_s$
15:    **return** $\mathcal{L}_{reg}, C_f, C_s, c_o, c_f, c_s$
16: **end function**

---

thresholding for HC. The algorithm first calculates complexity-driven scaling factors for confidence thresholds at the family and species levels (lines 2-3), ensuring that threshold values adapt to the complexity of the input data. This dynamic thresholding prevents overconfident predictions in ambiguous conditions. The core innovation lies in the taxonomy-aware fallback strategy (lines 4-8), where predictions with low confidence at fine-grained levels (e.g., species) can revert to higher-level categories (e.g., family or order). This approach significantly enhances model robustness, particularly in scenarios involving degraded images or ambiguous visual features, by maintaining semantic consistency in hierarchical predictions.

---

**Algorithm 3** Logic-driven Hierarchical Fallback Optimization

---

**Input**: Base thresholds $\theta_{f,0}, \theta_{s,0}$, Complexity coefficients $\gamma_f, \gamma_s$,
   Feature complexity scores $C_f, C_s$, Target confidences $c_o, c_f, c_s$,
   Level labels $\mathbf{Y}_l$, Order labels $\mathbf{Y}_o$, Family labels $\mathbf{Y}_f$, Species labels $\mathbf{Y}_s$

**Output**: Updated thresholds $\theta_f, \theta_s$, Adjusted prediction targets $\hat{Y}_f, \hat{Y}_s$

---

1: **function** DYNAMIC CONFIDENCE WITH FALLBACK($\theta_{f,0}, \theta_{s,0}, C_f, C_s, \gamma_f, \gamma_s, c_o, c_f, c_s, \mathbf{Y}_l, \mathbf{Y}_o, \mathbf{Y}_f, \mathbf{Y}_s$)
2:    $F \leftarrow [1.0 + \gamma_f \cdot C_f, 1.0 + \gamma_s \cdot C_s]$         ▷ Family and Species factors
3:    $\theta \leftarrow [\min(\max(\theta_{f,0} \cdot F[0], 0.5), 0.85), \min(\max(\theta_{s,0} \cdot F[1], 0.5), 0.85)]$   ▷ Thresholds for Family and Species levels
4:    $\hat{\mathbf{Y}} \leftarrow [\mathbf{Y}_f.\text{clone}(), \mathbf{Y}_s.\text{clone}()]$
5:    $\text{mask}_f \leftarrow (\mathbf{Y}_l == 2) \& (c_f < \theta[0])$         ▷ Low confidence at Family level
6:    $\text{mask}_s \leftarrow (\mathbf{Y}_l == 3) \& (c_s < \theta[1])$         ▷ Low confidence at Species level
7:    $\hat{\mathbf{Y}}[0][\text{mask}_f] \leftarrow \mathbf{Y}_o[\text{mask}_f]$         ▷ Fallback: Family → Order
8:    $\hat{\mathbf{Y}}[1][\text{mask}_s] \leftarrow \mathbf{Y}_f[\text{mask}_s]$         ▷ Fallback: Species → Family
9:    **return** $\theta[0], \theta[1], \hat{\mathbf{Y}}[0], \hat{\mathbf{Y}}[1]$         ▷ Updated $\theta_f, \theta_s$, Fallback targets $\hat{Y}_f, \hat{Y}_s$
10: **end function**

---

Effective feature selection is critical for accurate classification in complex visual scenarios. Algorithm 4 introduces an Attention Enhancement module driven by Position Prompts, which optimizes the model's focus on semantically important regions. The algorithm processes multi-head attention matrices from transformer layers (lines 3-9), employing a novel minimum-attention filtering strategy to identify consistently important regions across layers. The primary innovation lies in the generation of position prompts (lines 10-14), which incorporate class frequency statistics to mitigate class imbalance, ensuring that rare or visually similar species receive adequate attention. The top-K patch selection (lines 15-16) further refines focus, providing targeted guidance for the model's atten-

tion mechanism. This strategy significantly improves model performance in distinguishing visually similar species, where fine-grained morphological differences are critical.

---

**Algorithm 4** Attention Enhancement with Position Prompts

---

**Input**: Attention matrices $\mathbf{A}$ from transformer layers, Species labels $\mathbf{Y}_s$,
  Number of species classes $C_{\text{species}}$

**Output**: Position prompts $\mathbf{P}$, Attention loss $\mathcal{L}_{\text{att}}$

1: **function** ATTENTION ENHANCEMENT WITH POSITION PROMPTS($\mathbf{A}, \mathbf{Y}_s, C_{\text{species}}$)
2: $\quad \mathbf{I} \leftarrow \mathbf{I}_{N_{\text{patches}}}$ $\qquad\qquad\qquad\qquad\qquad\qquad\qquad\qquad\triangleright$ Initialize with identity matrix (rollout start)
3: $\quad$ **for** $l \leftarrow 1$ to $L$ **do**
4: $\qquad \bar{\mathbf{A}}_l \leftarrow \text{mean}(\mathbf{A}[l], \dim = 1)[:, 1:, 1:]$ $\qquad\qquad\triangleright$ Mean attention across heads, exclude CLS
5: $\qquad \mathbf{I} \leftarrow \left(\alpha\bar{\mathbf{A}}_l + (1-\alpha)\mathbf{I}_{N_{\text{patches}}}\right) \cdot \mathbf{I}$ $\quad\triangleright$ Propagate with residual identity mixing (controlled by $\alpha$)
6: $\quad$ **end for**
7: $\quad \text{imp} \leftarrow \text{mean}(\mathbf{I}, \dim = 1)$ $\qquad\qquad\qquad\qquad\qquad\qquad\triangleright$ Average importance per patch
8: $\quad \text{freq} \leftarrow \text{bincount}(\mathbf{Y}_s, \text{minlength} = C_{\text{species}})$ $\qquad\qquad\triangleright$ Class frequency statistics
9: $\quad \mathbf{W} \leftarrow \frac{(1/\max(\text{freq},1))^{0.5}}{\sum((1/\text{freq})^{0.5})} \cdot C_{\text{species}}$ $\qquad\qquad\triangleright$ Inverse sqrt frequency weights
10: $\quad \text{topk\_idx} \leftarrow \text{argsort}(\text{imp}, \dim = -1, \text{descending=True})[:, :K]$ $\qquad\triangleright$ K important patches
11: $\quad \mathbf{P} \leftarrow \text{scatter}(\mathbf{0}, \dim = 1, \text{index} = \text{topk\_idx}, \text{value} = 1)$ $\qquad\triangleright$ Create prompt mask
12: $\quad \mathbf{P} \leftarrow \mathbf{P} \cdot \text{imp} \cdot \mathbf{W}[\mathbf{Y}_s].\text{view}(-1, 1)$ $\qquad\qquad\triangleright$ Apply importance and class weights
13: $\quad \mathbf{P} \leftarrow \mathbf{P} \cdot \frac{K}{\sum(\mathbf{P}, \dim=1, \text{keepdim=True})}$ $\qquad\qquad\qquad\triangleright$ Normalize prompt weights
14: $\quad \mathbf{A}_L \leftarrow \mathbf{A}[-1].\text{mean}(1)[:, 0, 1:]$ $\qquad\qquad\qquad\triangleright$ CLS token's attention to patches
15: $\quad \mathcal{L}_{\text{att}} \leftarrow -\text{mean}\left(\sum \mathbf{P} \cdot \log \text{softmax}(\mathbf{A}_L, \dim = -1)\right)$ $\qquad\triangleright$ Cross-entropy loss
16: $\quad$ **return** $\mathbf{P}, \mathcal{L}_{\text{att}}$
17: **end function**

---

## A.4 OPTIMIZATION DYNAMICS AND EXTENDED ANALYSIS

This section provides gradient-based optimization analyses that are intended to offer conceptual insight into how the proposed regularizers and decision rules influence the training dynamics of LRConfNet. Following common practice in deep learning The goal is interpretative: to clarify why the LogReg term encourages parent–child consistency and how the uncertainty-aware thresholds interact with the hierarchical fallback strategy, not to establish new theoretical guarantees.

### A.4.1 MULTI-MODAL FEATURE EXTRACTION WITH SEMANTIC GUIDANCE

This section provides a detailed theoretical analysis of the ViT and Semantic-Guided Cross-Attention (SGCA) mechanisms. The complete mathematical derivation of SGCA, including the formulation of cross-attention and semantic fusion, is presented here for clarity.

### A.4.2 FORWARD PROPAGATION OF VISION TRANSFORMER

Starting with the input image $X \in \mathbb{R}^{H \times W \times C}$, it is divided into $N$ non-overlapping patches, each encoded as:

$$\mathbf{Z}_0 = \left[\mathbf{E}_{\text{patch}}^1, \mathbf{E}_{\text{patch}}^2, \dots, \mathbf{E}_{\text{patch}}^N\right] + \mathbf{E}_{\text{pos}} \tag{15}$$

The detailed propagation through each Transformer layer is defined as:

$$\mathbf{Z}_k' = \text{MSA}\left(\text{LN}\left(\mathbf{Z}_{k-1}\right)\right) + \mathbf{Z}_{k-1} \tag{16}$$

$$\mathbf{Z}_k = \text{MLP}\left(\text{LN}\left(\mathbf{Z}_k'\right)\right) + \mathbf{Z}_k' \tag{17}$$

The final visual representation after $K$ Transformer layers is denoted as $\mathbf{Z}_K$.

### A.4.3 SEMANTIC-GUIDED CROSS-ATTENTION MODULE

We further derive the cross-attention mechanism of SGCA, where semantic vectors $\mathbf{V}$ serve as queries ($Q$), and the visual features $\mathbf{Z}_L$ as keys ($K$) and values ($V$):

$$\mathbf{W} = \mathrm{softmax}\left(\frac{QK^T}{\sqrt{d}}\right), \quad \mathbf{F} = \mathbf{W}V \qquad (18)$$

The refined semantic-visual features $\mathbf{F}_M$ are produced through multi-layer cross-attention, enhancing the semantic alignment of visual features for HC.

## A.5 LOGICAL REASONING REGULARIZATION

LogReg is formulated based on the principle of hierarchical consistency, ensuring that predictions respect the logical structure of the taxonomy. We begin with the hierarchical consistency constraint:

$$\mathcal{L}_c = \mathbb{E}_{(X,Y)\sim\mathcal{D}}\left[\sum_{l=2}^{L} \lambda_{l,l-1} \cdot \mathrm{ReLU}\left(P_l(y_l \mid X) - P_{l-1}(y_{l-1} \mid X)\right)\right] \qquad (19)$$

The derivation begins with the probability consistency constraint:

$$P_{l-1} \geq P_l \qquad (20)$$

This is enforced using the ReLU function, ensuring that any violation is penalized. The hyperparameters $\lambda_{l,l-1}$ control the strength of this regularization.

### A.5.1 FEATURE CONSISTENCY

In addition to probability alignment, we ensure consistency in the feature space:

$$\mathcal{L}_s = \mathbb{E}_{X\sim\mathcal{D}}\Big[\sum_{l=2}^{L} \bar{\lambda}_{l-1} \cdot \left(1 - \frac{\bar{F}_l \cdot \bar{F}_{l-1}}{||\bar{F}_l||\,||\bar{F}_{l-1}||}\right)\Big] \qquad (21)$$

The derivation explores how cosine similarity can be used to enforce geometric alignment between adjacent level features.

### A.5.2 UNCERTAINTY QUANTIFICATION

Our UQ mechanism quantifies feature complexity through two metrics: (i) Feature Entropy and (ii) Feature Variance.

Feature Entropy:

$$\mathcal{H}_{\mathrm{sample}}(f'_l) = -\sum_{i=1}^{D_l} f'_{l,i} \log(\max(f'_{l,i}, \epsilon)) \qquad (22)$$

Feature Variance:

$$\sigma^2(f'_l) = \frac{1}{D_l} \sum_{i=1}^{D_l} (f'_{l,i} - \mu_{f'_l})^2 \qquad (23)$$

These two metrics are combined to form the complexity measure:

$$C(f'_l) = \mathcal{H}_{\mathrm{sample}}(f'_l) + \lambda_{\mathrm{var}} \cdot \sigma^2(f'_l) \qquad (24)$$

The derivation further explores the relationship between these metrics and model uncertainty, highlighting how they guide the adaptive confidence adjustment mechanism of LRConfNet.

## A.6 DYNAMIC CONFIDENCE ADJUSTMENT

We provide the full derivation of the logic-driven dynamic confidence adjustment framework. Starting from a Bayesian decision-theoretic perspective (Wu & Cameron, 1990), the decision at each level $l$ is to either accept the predicted class $Y_l$ or fallback to the parent category $Y_{l-1}$, minimizing the expected hierarchical loss:

$$a^* = \arg\min_a \mathbb{E}_{y_{\mathrm{true}}|X}[\mathcal{L}(a, y_{\mathrm{true}})] = \arg\min_a \sum_{y'} \mathcal{L}(a, y')P(y' \mid X) \qquad (25)$$

Under a binary loss setting: - $\mathcal{L}(a_l, y_l) = 0$ if correct, 1 otherwise. - $\mathcal{L}(a_{l-1}, y_l) = \delta_{l,l-1} \geq 0$, representing the cost of a fallback.

The optimal decision threshold is derived as:

$$p_l \geq \frac{\delta_{l,l-1}}{\delta_{l,l-1} + 1} \tag{26}$$

However, since feature complexity $C_l$ affects the reliability of $p_l$, we adapt this rule to:

$$\theta_l^{(e)} = \min\left(\max\left(\theta_{l,0} \cdot (1 + \gamma_l \cdot C_l^{(e)}), \theta_{\min}\right), \theta_{\max}\right) \tag{27}$$

This adjustment scales the base threshold by a complexity penalty factor, making it more stringent under high feature uncertainty.

### A.7 HIERARCHICAL FALLBACK AND SEMANTIC CONSISTENCY

The fallback mechanism operates directly based on these dynamic thresholds. When $p_l < \theta_l^{(e)}$, the model reverts to the higher-level category:

$$\hat{Y}_l = \begin{cases} Y_{\text{parent}(l)} & \text{if } p_l < \theta_l^{(e)} \\ Y_l & \text{if } p_l \geq \theta_l^{(e)} \end{cases} \tag{28}$$

The semantic consistency of this fallback is further ensured by the feature similarity constraint:

$$\mathcal{L}_s = \mathbb{E}_{X \sim D}\Big[\sum_{l=2}^{L} \bar{\lambda}_{l-1} \cdot \Big(1 - \frac{\bar{F}_l \cdot \bar{F}_{l-1}}{||\bar{F}_l|| \, ||\bar{F}_{l-1}||}\Big)\Big] \tag{29}$$

This aligns hierarchical predictions even when fine-grained details are unreliable.

### A.8 MULTI-LEVEL LOSS OPTIMIZATION STRATEGY WITH POSITION-AWARE ATTENTION ENHANCEMENT

#### A.8.1 RISK-AWARE DECISION ANALYSIS

Our approach is further justified as a risk-aware strategy, balancing fine-grained accuracy with hierarchical consistency. This is particularly important for degraded images, where fine-grained features are often unreliable, but higher-level semantic categories remain consistent.

#### A.8.2 MULTI-LAYER ATTENTION INTEGRATION

The cumulative attention map $\mathcal{R}$ in Eq. equation 8 is derived using the attention rollout technique (Abnar & Zuidema, 2020), which aggregates attention across multiple transformer layers. Formally, each attention matrix $\bar{\mathcal{A}}_l$ is calculated as:

$$\bar{\mathcal{A}}_l = \frac{1}{H} \sum_{h=1}^{H} \text{softmax}\left(\frac{Q_l^h K_l^{hT}}{\sqrt{d_k}}\right) \tag{30}$$

where $H$ is the number of attention heads, and $Q_l, K_l$ denote query and key matrices at layer $l$. This formulation ensures that the attention scores across heads are properly normalized.

#### A.8.3 CLASS-BALANCED ATTENTION WEIGHTING

Our class-balanced weighting in Eq. equation 9 is derived from the inverse frequency weighting principle (Cui et al., 2019). This approach minimizes a balanced empirical risk:

$$\hat{\mathcal{R}}(f) = \frac{1}{|D|} \sum_{(x,y) \in D} w_y \mathcal{L}(f(x), y) \tag{31}$$

This ensures that rare classes receive greater consideration, preventing them from being overshadowed by dominant classes.

Table 5: Ablation Study on the Effectiveness of SGCA, LogReg, and PAAE Modules on HRSC-Deg and CUB-Deg

| Methods | | | HRSC-Deg | | | | CUB-Deg |
|---|---|---|---|---|---|---|---|
| SGCA | LogReg | PAAE | *Fine Acc.* | SGCA | LogReg | PAAE | *Species Acc.* |
| ✗ | ✗ | ✗ | 87.50 | ✗ | ✗ | ✗ | 81.50 |
| ✓ | ✗ | ✗ | 89.40 | ✓ | ✗ | ✗ | 83.26 |
| ✗ | ✗ | ✓ | 88.04 | ✗ | ✗ | ✓ | 83.06 |
| ✓ | ✗ | ✓ | 89.94 | ✓ | ✗ | ✓ | 84.82 |
| ✗ | ✓ | ✗ | 87.59 | ✗ | ✓ | ✗ | 82.95 |
| ✓ | ✓ | ✗ | 89.49 | ✓ | ✓ | ✗ | 84.71 |
| ✗ | ✓ | ✓ | 89.86 | ✗ | ✓ | ✓ | 83.27 |
| ✓ | ✓ | ✓ | **91.76** | ✓ | ✓ | ✓ | **85.03** |

### A.8.4 POSITION PROMPTING AND ATTENTION LOSS DERIVATION

We identify the set $K_i$ of indices corresponding to the top $K$ most salient patches for each sample $i$, based on $I_i$. This imposes sparsity and focuses computational effort. $M_{K,ij} = 1$ if $j \in K_i$, else 0.

The prompts are normalized to form a valid target distribution (summing to 1 or a constant $K$ for subsequent loss calculation):

$$P'_{ij} = \frac{P_{ij}}{\sum_{k=1}^{N} P_{ik} + \epsilon} \quad \text{or} \quad P'_{ij} = \frac{P_{ij}}{\sum_{k=1}^{N} P_{ik} + \epsilon} \cdot K \tag{32}$$

We use the latter normalization (sum to K) consistent with the implementation details. $P'$ represents the desired focus intensity for each patch.

The Position Prompting and Attention Loss (Eq. equation 11) is derived from the Kullback-Leibler (KL) divergence between the model's predicted attention distribution and the target position prompt distribution:

$$\mathcal{L}_{\text{att}} = D_{\text{KL}}(P' \| \text{softmax}(\mathcal{A}^{(L)})) \tag{33}$$

This formulation ensures that the model's focus aligns with the most informative regions identified by the Position Prompts, providing targeted guidance even under degraded conditions.

### A.9 MORE EXPERIMENTAL DETAILS AND RESULTS

#### A.9.1 MORE DETAILS AND RESULTS OF ABLATION EXPERIMENTS

**1) Effectiveness of SGCA, LogReg, and PAAE.** To further dissect the contribution of each proposed module, we conduct a comprehensive ablation study on SGCA (Semantic-Guided Cross-Attention), LogReg (Logical Reasoning Regularization), and PAAE (Position-Aware Attention Enhancement). The results in Table 5 clearly demonstrate the incremental benefits of each component. Removing all three modules leads to a strong degradation in fine-grained accuracy. Introducing SGCA alone improves accuracy by +1.90% on HRSC-Deg and +1.76% on CUB-Deg, validating the importance of semantic priors and cross-modal alignment in compensating for degraded visual features. PAAE further contributes by enhancing spatial localization, yielding +1.56% improvement on CUB-Deg, while LogReg ensures logical consistency across hierarchical levels, slightly boosting fine-level accuracy and preventing over-classification errors. The synergy of combining SGCA with PAAE achieves 89.94% and 84.82% on HRSC-Deg and CUB-Deg, respectively, highlighting their complementary roles in refining feature representation and region focus. The full model integrating all three modules achieves the best performance, which constitutes an overall improvement of +4.26% and +3.53% over the baseline. These results confirm that each module is indispensable: SGCA bridges semantic-visual gaps, PAAE sharpens degraded spatial features, and LogReg enforces hierarchical reasoning, together forming a coherent and robust framework for hierarchical classification under real-world degradations.

**2) LogReg Component Decomposition.** To further disentangle the contributions of LogReg, we conduct a component-wise ablation study by isolating its three core elements: *Hierarchical Consistency* (Eq. 1), *Feature Consistency* (Eq. 2), and the *Uncertainty Quantification* mechanism (Eq. 3–4).

Table 6: Ablation Study on LogReg Components (Hierarchical Consistency (HC)), Feature Consistency (FC), and Uncertainty Quantification (UQ)

| Configuration | HRSC-Deg | | | | CUB-Deg | | | |
|---|---|---|---|---|---|---|---|---|
| Setup | HC | FC | UQ | *Fine Acc.* | HC | FC | UQ | *Species Acc.* |
| Baseline | ✗ | ✗ | ✗ | 89.40 | ✗ | ✗ | ✗ | 83.26 |
| HC | ✓ | ✗ | ✗ | 89.59 | ✓ | ✗ | ✗ | 84.11 |
| FC | ✗ | ✓ | ✗ | 89.34 | ✗ | ✓ | ✗ | 83.79 |
| UQ | ✗ | ✗ | ✓ | 90.11 | ✗ | ✗ | ✓ | 84.27 |
| HC+FC | ✓ | ✓ | ✗ | 89.77 | ✓ | ✓ | ✗ | 84.15 |
| Full LogReg | ✓ | ✓ | ✓ | **91.76** | ✓ | ✓ | ✓ | **85.03** |

Table 7: Ablation Study of Attention and Gradient Fusion in PAAE (CUB-Deg)

| Component | Attention Only | Gradient Only | Fusion ($\gamma = 0.5$) | *Species Acc.* |
|---|---|---|---|---|
| None | ✗ | ✗ | ✗ | 83.70 |
| Attention | ✓ | ✗ | ✗ | 85.10 |
| Gradient | ✗ | ✓ | ✓ | 84.80 |
| Fusion | ✗ | ✗ | ✓ | 86.90 |

As shown in Table 6, each component provides complementary benefits. On HRSC-Deg, Hierarchical Consistency alone leads to a slight improvement in fine-grained accuracy, confirming the importance of enforcing logical parent–child constraints. Feature Consistency achieves comparable performance, validating the role of geometric alignment in the feature space. The UQ mechanism yields a larger gain, reflecting its effectiveness in mitigating unreliable predictions under degraded conditions. Combining Hierarchical and Feature Consistency further stabilizes predictions, while the full LogReg integration achieves the highest performance. These results clearly demonstrate that all three components are indispensable, with UQ providing robustness to noise and Feature Consistency reinforcing semantic alignment, while their integration ensures a logically consistent and uncertainty-aware hierarchical classifier.

**3) PAAE Component Ablation.** To further validate the necessity of each design within PAAE, we perform fine-grained ablations on its three core components: importance computation, position prompting, and class balance weighting. As shown in Table 8, removing any component consistently degrades performance. Specifically, excluding the importance computation module reduces accuracy by 1.33%, highlighting the contribution of attention-gradient fusion to reliable salience estimation. Similarly, removing position prompts decreases accuracy by 1.23%, confirming their role in guiding focus toward semantically critical regions. Class balance weighting yields the most substantial drop, demonstrating its effectiveness in mitigating long-tail imbalance. We also tested simplified settings (attention-only and gradient-only), both of which underperform the full fusion strategy, confirming that complementary cues are necessary. Altogether, these results underscore that each submodule is indispensable and that their integration yields the strongest robustness under degraded conditions.

**4) Effectiveness of Attention-Gradient Fusion**. To verify the necessity of attention-gradient fusion in PAAE, we compare attention-only, gradient-only, and their fusion variants on CUB-Deg. As shown in Table 7, both attention and gradient cues individually improve species accuracy over the baseline, confirming that each provides complementary salience information. However, their simple fusion ($\gamma = 0.5$) yields the largest gain, boosting accuracy to 86.90%, which is +3.2% higher than the baseline. This demonstrates that combining global attention priors with gradient-based local sensitivity produces more stable and discriminative importance maps, effectively guiding the model under degraded conditions. These results confirm that attention-gradient fusion is not only beneficial but essential to the robustness of LRConfNet.

Table 8: Ablation Study of PAAE Components on CUB-Deg

| Methods | CUB-Deg | |
| --- | --- | --- |
| Config | *Species Acc.* | *Δ Perf.* |
| Full PAAE | **85.03** | - |
| w/o Importance | 83.70 | -1.33% |
| w/o Position Prompts | 83.80 | -1.23% |
| w/o Class Balance | 83.26 | -1.77% |
| Attention Only | 84.10 | -0.93% |
| Gradient Only | 83.80 | -1.23% |

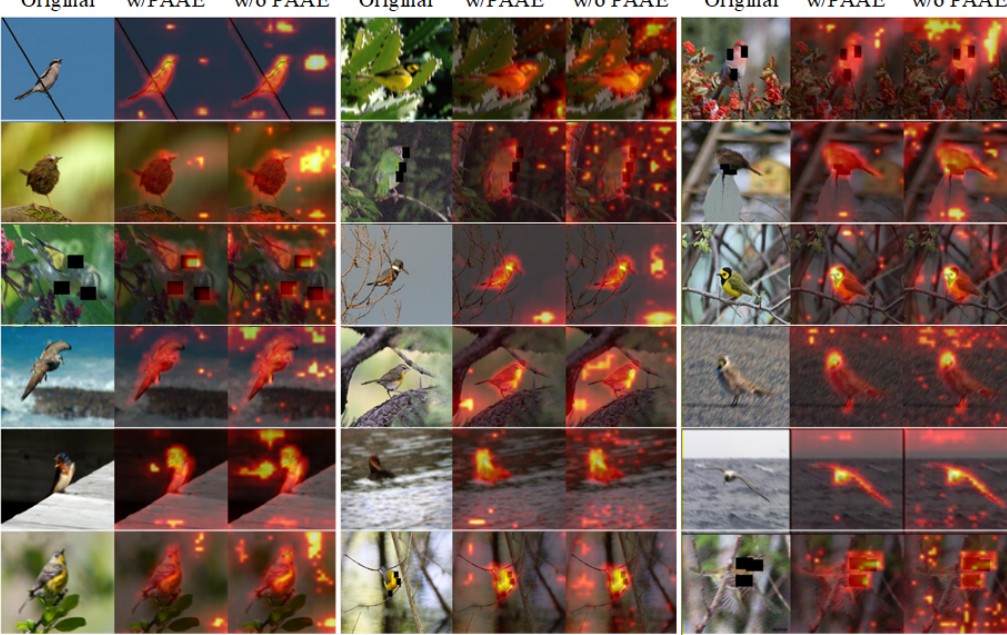

Figure 6: Qualitative attention-rollout comparison of the proposed PAAE module on CUB-Deg. Each triplet shows the original degraded image (left), the attention map with PAAE (middle), and the attention map without PAAE (right). Across diverse degradations (blur, noise, low resolution, and occlusion) and long-tailed categories, PAAE consistently concentrates attention on semantically discriminative bird parts (e.g., head, wings, and body contours), whereas the baseline often focuses on background clutter or spurious artifacts. These visualizations demonstrate that PAAE supplies more reliable fine-grained evidence under degradation, which in turn supports the risk-aware, semantically consistent hierarchical decisions made by LRConfNet.

### A.9.2 ADDITIONAL QUALITATIVE VISUALIZATIONS FOR PAAE

Figure 6 provides qualitative attention-rollout visualizations on CUB-Deg, comparing LRConfNet with and without the proposed PAAE module. For each example, we show the original degraded image, the attention map with PAAE, and the attention map without PAAE. Under various degradation types (blur, noise, low resolution and partial occlusion), PAAE consistently drives the model's focus toward semantically discriminative bird regions (e.g., head, wings, and body contours), while the baseline without PAAE often attends to background clutter or spurious artifacts. This confirms that PAAE supplies more reliable fine-grained visual evidence under degradation, which is then exploited by LogReg and the dynamic fallback mechanism to support risk-aware, semantically consistent hierarchical decisions, especially for rare and hard examples.

Table 9: Impact of Confidence Thresholds on Classification Performance for HRSC-Deg Dataset

| $T_f$ | $T_c$ | $ISDL$ | $P_H$ | $R_H$ | $Lvl_{acc}$ | Class_acc | |
|---|---|---|---|---|---|---|---|
| | | | | | | Coarse | Fine |
| 0.73 | 0.58 | 70.54 | 91.76 | 88.85 | 74.51 | 98.51 | 84.84 |
| 0.73 | 0.60 | 69.79 | 91.43 | 88.13 | 73.74 | 99.40 | 83.80 |
| 0.73 | 0.62 | 73.03 | 92.53 | 89.67 | 74.18 | 99.11 | 83.45 |
| 0.75 | 0.62 | 70.60 | 90.44 | 88.74 | 74.29 | 98.51 | 85.02 |
| 0.75 | 0.58 | 71.48 | 91.15 | 89.40 | 74.51 | 98.51 | 84.49 |
| 0.75 | 0.60 | 70.00 | 90.16 | 88.74 | 74.95 | 98.21 | 83.80 |
| 0.77 | 0.58 | 69.41 | 90.27 | 88.52 | 74.29 | 98.21 | 85.54 |
| 0.77 | 0.60 | 70.65 | 90.49 | 89.29 | 74.62 | 99.11 | 83.80 |
| 0.77 | 0.62 | 68.68 | 90.77 | 88.57 | 73.08 | 98.51 | 81.88 |

Note: $T_f$ and $T_c$ denote fine-grained and coarse-grained confidence thresholds, respectively. Other model parameters were fixed during this sensitivity analysis: $\alpha_{unc} = 0.04$, $\beta_{decay} = 0.3$, $b_{0,conf} = 0.65$, $\omega_{reg} = 0.1$, $\gamma_{complex} = 0.03$, $\lambda_{attn} = 0.5$, $\tau = 2$, $k = 14$, and $\eta = 0.6$. $ISDL$ measures the structural consistency between prediction and ground truth, $P_H$ and $R_H$ denote Hierarchical Precision and Recall, $Lvl_{acc}$ refers to Level Accuracy. All metrics are reported as percentages (%).

### A.9.3 SENSITIVITY ANALYSIS OF PARAMETERS

We conducted a comprehensive sensitivity analysis of LRConfNet's key parameters, including confidence thresholds ($T_f$, $T_c$), dynamic confidence adjustment factors ($\alpha_{unc}$, $\beta_{decay}$, $b_{0,conf}$), PAAE parameters ($\lambda_{attn}$, $\tau$, $k$, $\rho$), and LogReg parameters ($\omega_{reg}$, $\gamma_{family}$, $\gamma_{species}$). Our analysis reveals that optimal configurations differ across metrics, with fine-grained classification accuracy benefiting from higher fine-grained thresholds, while balanced hierarchical performance is achieved with moderate threshold combinations. Lower values of $\alpha_{unc}$ and moderate $\beta_{decay}$ settings enhance $P_H$ and ISDL, confirming the importance of adaptive uncertainty calibration. PAAE parameters exhibit distinct optimal settings across hierarchical levels, with each level benefiting from tailored attention configurations. Additionally, LogReg demonstrates the best performance under moderate regularization and appropriately scaled complexity coefficients for family and species levels, ensuring semantic consistency without overly constraining predictions. These results validate LRConfNet's adaptability to varying classification objectives, highlighting the importance of calibrated parameter configurations in maintaining robust HC under degraded conditions. $T_{fine}$ and $T_{coarse}$ denote fine- and coarse-level decision thresholds; for three-level settings we use $T_{order}$, $T_{family}$, and $T_{species}$. Dynamic clipping uses $\theta_{min}$ and $\theta_{max}$.

To understand how confidence thresholds impact model performance, Table 9 illustrates the sensitivity of our dynamic confidence adjustment framework to threshold parameters ($T_f$ and $T_c$) on the HRSC-Deg dataset. Our analysis reveals distinct optimal configurations for different performance objectives: ISDL and $P_H$ maximize with moderate fine-grained thresholds and higher coarse-grained thresholds, while fine-grained accuracy benefits from higher $T_f$ values paired with lower $T_c$ settings. Level accuracy achieves optimal results with balanced threshold combinations. These divergent optima highlight how confidence thresholds regulate the uncertainty-driven decision refinement process, determining when the model should trust fine-grained predictions versus triggering logic-driven hierarchical fallbacks. The framework's ability to be calibrated for different hierarchical priorities demonstrates its adaptability in handling degraded remote sensing imagery, where visual feature degradation makes confidence assessment and logical consistency essential. The observed relationship between threshold configurations and hierarchical metrics validates our approach's effectiveness in maintaining semantic coherence while navigating the complexity-accuracy trade-off inherent in HC of degraded images.

Our investigation extends to the parameters governing UQ, as shown in Table 10. This table examines the sensitivity of our UQ mechanism to key dynamic confidence adjustment parameters: uncertainty rate ($\alpha_{unc}$), confidence decay factor ($\beta_{decay}$), and initial confidence baseline ($b_{0,conf}$) on the HRSC-Deg dataset. Our analysis reveals that lower $\alpha$ values combined with moderate $\beta$ settings yield optimal performance across hierarchical metrics, particularly for ISDL and hierarchical precision. This configuration enables more cautious uncertainty estimation, allowing the model to better identify when fine-grained predictions may be unreliable in degraded remote sensing imagery. No-

Table 10: Sensitivity Analysis of Dynamic Confidence Adjustment Parameters on HRSC-Deg Dataset

| $\alpha_{\text{unc}}$ | $\beta_{\text{decay}}$ | $b_{0,\text{conf}}$ | $ISDL$ | $P_H$ | $R_H$ | $Lvl_{acc}$ | Class_acc | |
|---|---|---|---|---|---|---|---|---|
| | | | | | | | Coarse | Fine |
| 0.03 | 0.3 | 0.63 | 72.39 | 91.32 | 89.62 | 74.95 | 98.81 | 86.41 |
| 0.03 | 0.4 | 0.63 | 70.76 | 91.70 | 89.01 | 73.63 | 98.51 | 83.80 |
| 0.03 | 0.3 | 0.67 | 70.65 | 91.59 | 88.24 | 72.86 | 98.51 | 82.75 |
| 0.03 | 0.4 | 0.67 | 69.95 | 90.49 | 89.07 | 74.40 | 97.62 | 82.75 |
| 0.04 | 0.4 | 0.67 | 70.71 | 91.32 | 89.18 | 73.85 | 98.81 | 84.67 |
| 0.04 | 0.3 | 0.63 | 70.54 | 90.93 | 89.07 | 74.07 | 97.62 | 83.28 |
| 0.04 | 0.3 | 0.67 | 69.79 | 90.38 | 88.52 | 73.74 | 98.51 | 83.28 |
| 0.04 | 0.4 | 0.63 | 70.98 | 91.92 | 89.73 | 73.96 | 98.21 | 82.40 |

Note: $\alpha_{\text{unc}}$ is the uncertainty-rate parameter, $\beta_{\text{decay}}$ the confidence-decay factor, and $b_{0,\text{conf}}$ the initial confidence baseline. Other model parameters were fixed: $T_f = 0.77$, $T_c = 0.62$, $\omega_{reg} = 0.1$, $\gamma_{complex} = 0.03$, $\lambda_{attn} = 0.5$, $\tau = 2$, $k = 14$, and $\eta = 0.6$. $ISDL$ measures the structural consistency between prediction and ground truth, $P_H$ and $R_H$ denote Hierarchical Precision and Recall. All metrics are reported as percentages (%).

tably, the initial confidence baseline ($b_{0,\text{conf}}$) significantly impacts the balance between coarse and fine-grained accuracy, with lower values generally favoring hierarchical consistency. These findings confirm the efficacy of our dynamic confidence adjustment framework, which adaptively refines classification decisions based on estimated uncertainty levels while maintaining semantic coherence through logical reasoning constraints. The observed parameter sensitivities validate our approach to UQ, demonstrating how carefully calibrated confidence dynamics can guide the hierarchical fallback mechanism to achieve robust classification even when visual features are compromised by image degradation. This adaptive uncertainty-driven behavior is particularly valuable for remote sensing applications where environmental conditions often affect image quality and where classification errors at fine-grained levels can propagate through the hierarchy.

We further explore the role of attention enhancement, with Table 11 analyzing the sensitivity of our PAAE mechanism to four key parameters: attention weight ($\lambda_{attn}$), temperature ($\tau$), top-k attention regions ($k$), and class balance power ($\rho$) on the CUB-Deg dataset. Our analysis reveals nuanced interactions between these parameters across the three-level hierarchical structure of degraded natural images. Configurations balancing moderate attention weights with appropriate temperature settings demonstrate superior performance in harmonizing classification across all taxonomic levels. Notably, the top-k parameter, which controls attention localization granularity, exhibits different optimal values for different hierarchy levels—smaller values benefit order-level recognition, while larger values enhance fine-grained species discrimination. The class balance power parameter proves particularly influential for addressing the inherent taxonomic imbalance in the CUB hierarchy, with moderate values yielding the best overall hierarchical consistency. These findings validate our attention enhancement strategy's effectiveness in focusing on discriminative regions despite image degradation, particularly for complex three-level classification tasks. The parameter sensitivity patterns confirm that our position-aware attention mechanism successfully adapts to the varying visual complexity across hierarchy levels, with attention parameters dynamically modulating feature emphasis based on classification difficulty. This adaptive attention allocation is essential for degraded natural images where fine-grained visual cues may be compromised, demonstrating how our framework maintains classification robustness by intelligently directing computational resources to informative regions while suppressing noise from degraded areas.

Logical consistency is a core aspect of our framework, and Table 12 explores the sensitivity of our Logical Reasoning Regularization (LogReg) mechanism to three critical parameters: regularization weight ($\omega_{reg}$) and complexity coefficients for family ($\gamma_{family}$) and species ($\gamma_{species}$) levels on the CUB-Deg dataset. Our analysis reveals distinctive patterns in how these parameters influence HC consistency across the three taxonomic levels of degraded natural images. Higher regularization weights combined with proportionally scaled complexity coefficients yield improved ISDL scores and level accuracy, demonstrating the effectiveness of stronger logical constraints in maintaining taxonomic consistency. Interestingly, the relationship between hierarchical precision ($P_H$) and recall ($R_H$) exhibits an inverse correlation as regularization strength increases, highlighting

Table 11: Sensitivity Analysis of PAAE Parameters on CUB-Deg Dataset

| $\lambda_{attn}$ | $\tau$ | $k$ | $\rho$ | $Lvl_{acc}$ | Class_acc | | | $ISDL$ | $P_H$ | $R_H$ |
|---|---|---|---|---|---|---|---|---|---|---|
| | | | | | Order | Family | Species | | | |
| 0.3 | 0.8 | 10 | 0.7 | 73.67 | 98.93 | 94.87 | 84.85 | 65.85 | 90.71 | 93.15 |
| 0.5 | 1 | 14 | 0.5 | 73.72 | 98.95 | 94.77 | 84.82 | 65.96 | 90.62 | 93.28 |
| 0.3 | 0.8 | 10 | 0.3 | 73.22 | 98.95 | 94.96 | 84.78 | 65.70 | 90.29 | 93.08 |
| 0.3 | 0.8 | 14 | 0.5 | 73.34 | 98.93 | 95.06 | 84.75 | 65.67 | 90.31 | 92.98 |
| 0.5 | 1 | 14 | 0.5 | 73.31 | 98.95 | 95.06 | 84.75 | 65.67 | 90.18 | 93.05 |
| 0.3 | 0.8 | 7 | 0.3 | 73.67 | 98.93 | 94.87 | 84.85 | 65.85 | 90.71 | 93.15 |
| 0.3 | 1.2 | 2 | 0.7 | 73.2 | 98.95 | 95.06 | 84.64 | 65.68 | 90.36 | 93.02 |

Note: $\lambda_{attn}$ represents attention weight, $\tau$ is the attention temperature parameter, $k$ denotes the number of top attention regions, and $\rho$ denotes the class-balance power coefficient (inverse-frequency exponent). Fixed parameters in this experiment: logical reasoning regularization (enabled), $\omega_{reg} = 0.05$, $\gamma_{family} = 0.03$, $\gamma_{species} = 0.04$, confidence thresholds ($T_{species} = 0.75$, $T_{family} = 0.7$), uncertainty parameters ($b_{0,species} = 0.65$, $b_{0,family} = 0.6$, $\alpha_{unc} = 0.04$, $\beta_{decay} = 0.3$). All metrics are reported as percentages (%).

Table 12: Sensitivity Analysis of Logical Reasoning Regularization Parameters on CUB-Deg Dataset

| $\omega_{reg}$ | $\gamma_{family}$ | $\gamma_{species}$ | $Lvl_{acc}$ | Class_acc | | | $ISDL$ | $P_H$ | $R_H$ |
|---|---|---|---|---|---|---|---|---|---|
| | | | | Order | Family | Species | | | |
| 0.03 | 0.02 | 0.03 | 73.05 | 98.95 | 94.49 | 84.85 | 65.33 | 89.22 | 93.58 |
| 0.05 | 0.03 | 0.04 | 73.34 | 98.93 | 95.06 | 84.75 | 65.67 | 90.31 | 92.98 |
| 0.08 | 0.04 | 0.05 | 73.76 | 98.95 | 94.87 | 84.78 | 66.31 | 90.48 | 92.97 |

Note: $\omega_{reg}$ represents the logical reasoning regularization weight, $\gamma_{family}$ and $\gamma_{species}$ denote complexity coefficients for family and species levels, respectively. These parameters control the strength and hierarchical balance of logical constraints in the regularization process. Fixed parameters in this experiment: attention parameters ($\lambda_{attn} = 0.3$, $\tau = 0.8$, $k = 14$), confidence thresholds ($T_{species} = 0.75$, $T_{family} = 0.7$), fallback thresholds ($T_{species\rightarrow family} = 0.68$, $T_{family\rightarrow order} = 0.63$), and uncertainty parameters ($b_{0,species} = 0.65$, $b_{0,family} = 0.6$, $\alpha_{unc} = 0.04$, $\beta_{decay} = 0.3$). All metrics are reported as percentages (%).

the fundamental trade-off between classification conservatism and coverage. Configurations with moderate regularization weights strike an optimal balance for the species level, while stronger regularization benefits intermediate levels where visual features may be more ambiguous in degraded images. These findings validate LogReg's crucial role in our framework, confirming that logical constraints effectively guide the hierarchical fallback mechanism when visual evidence is unreliable due to degradation. The sensitivity patterns further demonstrate how LogReg parameters can be calibrated to address class imbalance and hierarchical sparsity challenges in the CUB taxonomy, with complexity coefficients adaptively weighting the influence of logical rules at each level based on classification difficulty. This logical reasoning foundation provides semantic coherence to uncertainty-driven decisions, ensuring that classification outcomes respect taxonomic relationships even when processing degraded images with compromised visual features.

To explore how confidence thresholds impact classification decisions, Figure 7 provides a comprehensive visualization of confidence threshold sensitivity in our dynamic confidence adjustment framework, revealing distinct performance patterns across hierarchical metrics for HRSC-Deg degraded remote sensing images. The heatmap representations clearly illustrate how fine-grained ($T_f$) and coarse-grained ($T_c$) confidence thresholds interact to influence classification decisions at different hierarchy levels. Our analysis reveals unique optimal threshold configurations for each evaluation metric—Level Accuracy peaks at $T_f = 0.75$, $T_c = 0.60$, indicating a balanced hierarchical performance, while Fine-grained Accuracy maximizes at $T_f = 0.77$, $T_c = 0.58$, demonstrating the need for higher fine-grained thresholds to ensure classification precision at detailed levels. Conversely, Coarse-grained Accuracy achieves optimal results with $T_f = 0.73$, $T_c = 0.60$, suggesting that more conservative fine-grained thresholds better facilitate logical reasoning at broader taxo-

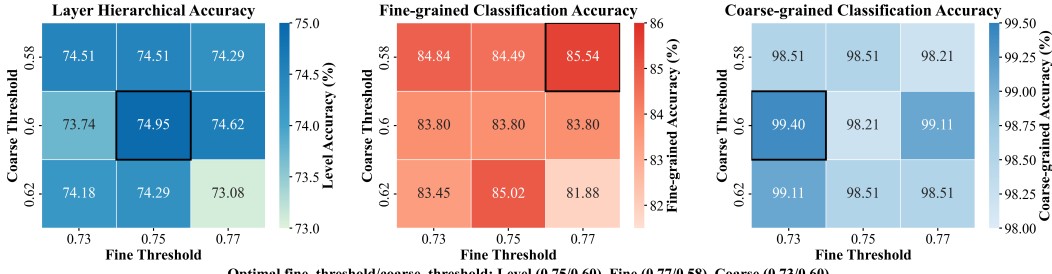

Figure 7: Confidence Threshold Sensitivity Analysis on HRSC-Deg Dataset: Impact on Hierarchical and Classification Performance Metrics. Heatmaps illustrating the impact of confidence thresholds on model performance across multiple metrics on the HRSC-Deg dataset. The x-axis ($T_f$) represents fine-grained confidence thresholds, while the y-axis ($T_c$) indicates coarse-grained confidence thresholds. Higher values are shown in darker colors within each heatmap.

nomic levels. These divergent optima highlight how our framework's UQ mechanism adaptively refines classification decisions based on confidence threshold calibration. The visible performance gradients across the heatmaps validate our logical reasoning-driven fallback strategy, which intelligently triggers hierarchical transitions when confidence falls below calibrated thresholds. This visual analysis confirms that optimal threshold configurations vary by objective, underscoring the importance of adaptive confidence adjustment in degraded remote sensing imagery where visual features may be compromised. The distinct performance patterns for hierarchical precision ($P_H$) and recall ($R_H$) further demonstrate how confidence thresholds modulate the balance between classification conservatism and coverage in our framework.

Our analysis of threshold interactions further extends in Figure 8, which illustrates the fine-grained interactions between confidence threshold parameters and multiple performance metrics in our dynamic confidence adjustment framework for degraded remote sensing imagery. The line plots reveal distinct sensitivity patterns across six critical metrics as fine-grained ($T_f$) and coarse-grained ($T_c$) thresholds vary, offering deeper insights into our uncertainty-driven decision refinement mechanism. Fine-grained Accuracy exhibits a striking non-monotonic relationship with threshold configurations, peaking at different combinations for different coarse thresholds, which validates our framework's ability to adaptively balance classification granularity. The ISDL metric shows remarkable sensitivity to higher coarse thresholds with lower fine thresholds, confirming the effectiveness of our logical reasoning constraints in maintaining hierarchical consistency under uncertainty. Notably, the hierarchical precision and recall curves demonstrate complementary behaviors, with $P_H$ generally decreasing as fine thresholds increase while $R_H$ shows more complex threshold interactions—highlighting how our confidence-driven fallback mechanism navigates the precision-recall trade-off in degraded imagery classification. Level Accuracy peaks with balanced threshold combinations, particularly at $T_f = 0.75$, $T_c = 0.60$, demonstrating how well-calibrated confidence parameters enable optimal hierarchical decision-making. The divergent trends across metrics underscore a fundamental aspect of our approach: confidence thresholds serve as key regulatory parameters in our UQ framework, determining when the model should trust fine-grained predictions versus triggering logic-driven hierarchical fallbacks. These visualizations confirm that optimal threshold configurations vary by objective, emphasizing the importance of adaptive confidence calibration in degraded remote sensing applications where visual features may be compromised.

Focusing on the role of attention mechanisms, Figure 9 presents a comprehensive analysis of our PAAE mechanism's sensitivity to four key parameters on the challenging CUB-Deg dataset: attention weight ($\lambda_{attn}$), temperature ($\tau$), top-k attention regions ($k$), and class balance power ($\rho$). The bar charts reveal how different PAAE configurations influence performance across the three-level hierarchical taxonomy of degraded natural images. Our analysis demonstrates that the optimal attention parameters vary significantly across hierarchical levels, validating our multi-level attention strategy's adaptability. Configuration C2 ($\lambda_{attn} = 0.5$, $\tau = 1.0$, $k = 14$, $\eta = 0.5$) achieves superior ISDL performance, indicating improved hierarchical consistency through balanced attention modulation. Notably, family-level accuracy benefits most from configurations with higher temperature

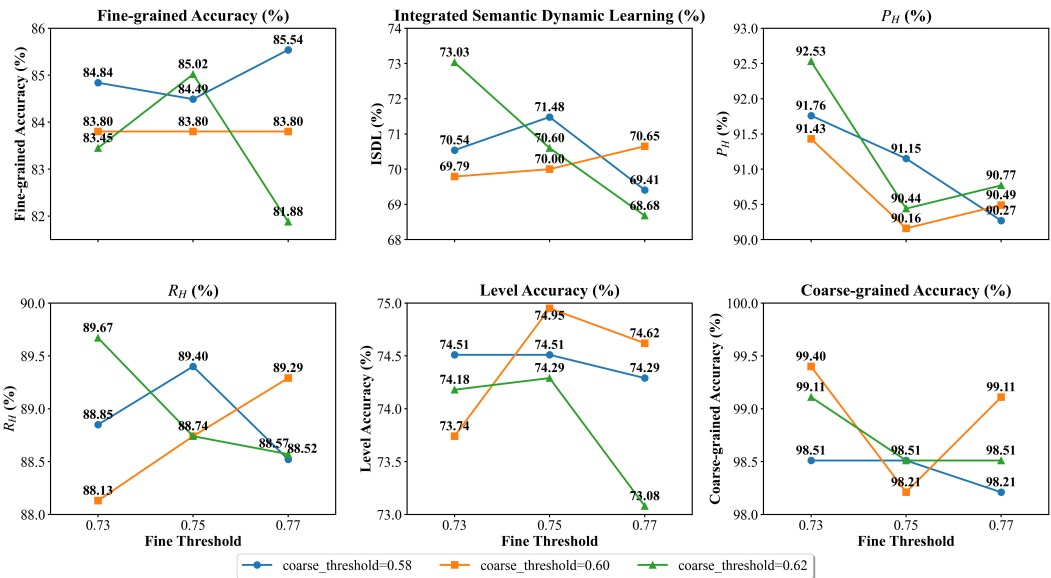

Figure 8: Multi-Metric Performance Analysis of Confidence Threshold Parameters on HRSC-Deg Dataset. Line plots showing the effect of varying confidence thresholds on six performance metrics for HC on the HRSC-Deg dataset. Each subplot represents a different evaluation metric plotted against the fine-grained threshold, with separate lines for three coarse-grained threshold values.

and moderate class balance values (C4, C5, C7), while species-level accuracy—the most challenging due to fine-grained distinctions in degraded images—is maximized by configurations with moderate attention weights and carefully calibrated top-k values (C1, C6). The divergent performance patterns across taxonomic levels confirm our position-aware approach's effectiveness in addressing the varying visual complexity and feature degradation at different hierarchy depths. The minimal variation in order-level accuracy across configurations, contrasted with significant differences at family and species levels, validates our framework's adaptive attention allocation strategy, which prioritizes computational resources toward informative regions based on classification difficulty and uncertainty. These findings demonstrate how our attention enhancement mechanism successfully navigates the challenges of degraded natural images where fine-grained visual cues may be compromised, contributing to the robustness of our logic-driven HC framework.

Logical consistency remains a critical aspect of our framework, as shown in Figure 10, which presents a systematic analysis of how Logical Reasoning Regularization (LogReg) parameters—regularization weight ($\omega_{reg}$) and complexity coefficients for family ($\gamma_{family}$) and species ($\gamma_{species}$) levels—influence our model's performance on the three-tiered CUB-Deg dataset. The bar charts reveal compelling trends across hierarchical metrics as regularization strength increases. Notably, Level Accuracy and ISDL demonstrate consistent improvements with stronger logical constraints (Config 3: $\omega_{reg} = 0.08$, $\gamma_{family} = 0.04$, $\gamma_{species} = 0.05$), confirming that reinforced logical consistency facilitates better overall HC in degraded natural images. Conversely, Species Accuracy peaks with lighter regularization (Config 1), while Family Accuracy is maximized with moderate regularization parameters (Config 2), revealing a fundamental level-specific response to logical constraints. Most revealing is the inverse relationship between hierarchical precision ($P_H$) and recall ($R_H$) across configurations—as regularization strengthens, $P_H$ increases while $R_H$ decreases, demonstrating how logical rules modulate the precision-recall trade-off within our uncertainty-driven framework. This phenomenon validates our logic-driven fallback strategy, which employs logical reasoning to determine when to trust fine-grained predictions versus triggering hierarchical fallbacks. The varying optimal configurations across different taxonomic levels underscore LogReg's critical role in balancing fine-grained discrimination against hierarchical consistency, especially crucial for degraded images where visual features may be compromised. These insights confirm that logical reasoning regularization provides essential semantic guidance to our dynamic

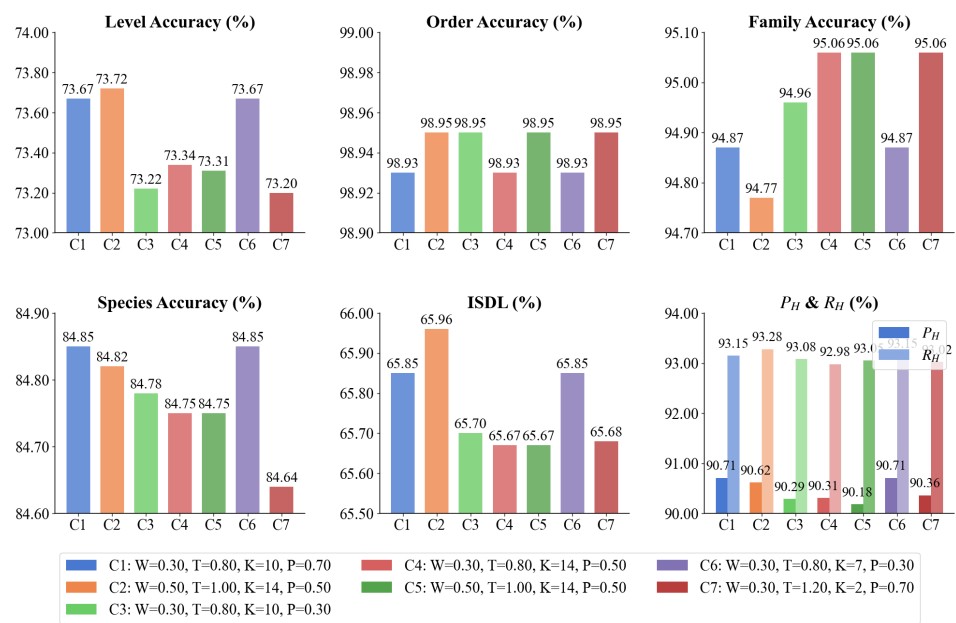

Figure 9: PAAE Parameter Analysis on CUB-Deg Dataset. Comparative analysis of PAAE parameter configurations on the CUB-Deg dataset. Six performance metrics are evaluated across seven parameter combinations (C1-C7), where each configuration varies in attention weight ($\lambda_{attn}$), temperature ($\tau$), top-k attention regions ($k$), and class balance power ($\rho$).

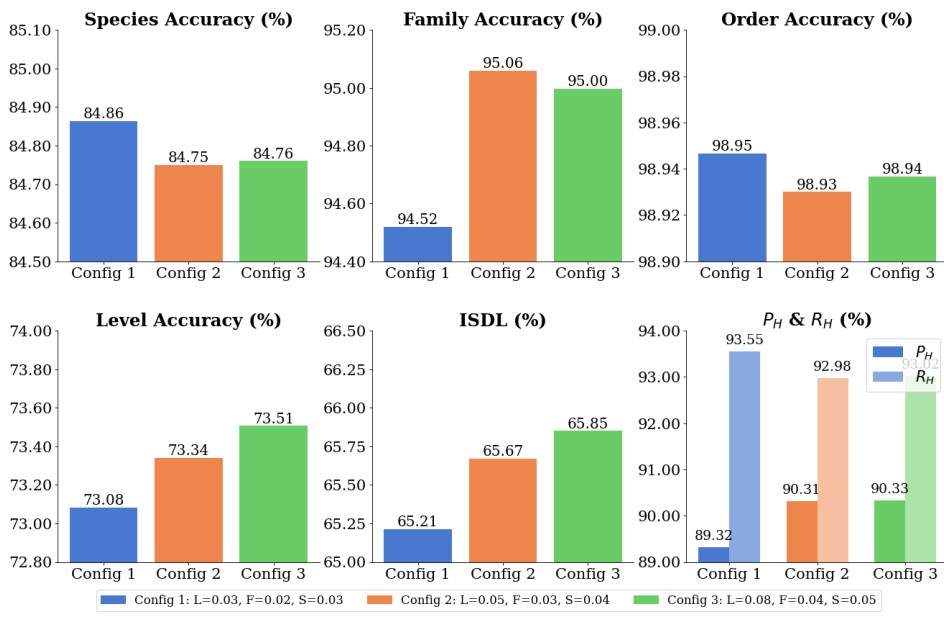

Figure 10: Analysis of Logical Reasoning Regularization Parameters on CUB-Deg Dataset. Evaluation of three logical reasoning regularization parameter configurations on the CUB-Deg dataset. Each configuration varies in regularization weight ($\omega_{reg}$) and complexity coefficients for family ($\gamma_{family}$) and species ($\gamma_{species}$) levels. Six performance metrics are presented across three increasingly stronger regularization settings.

confidence adjustment mechanism, enhancing decision reliability when classifying degraded natural images with complex three-level taxonomic relationships.

## B    USE OF LARGE LANGUAGE MODELS (LLMS)

We used a large language model (OpenAI GPT-5) only for minor language-related assistance. Specifically, the model was employed to:

- perform grammar and spelling correction,
- suggest small word or phrasing improvements to enhance readability and fluency.

The LLM was not involved in research ideation, methodology design, experiment execution, data analysis, or the generation of technical content. All scientific ideas, algorithms, results, and conclusions are entirely the responsibility of the authors.

