# OpenReview forum: "LRConfNet: Logical Reasoning-Driven Confidence Adjustment and Regularization for Hierarchical Classification of Degraded Images"
_ICLR.cc/2026/Conference — Submitted to ICLR 2026_

### Official Review · Reviewer_sRUs · 2025-10-25

**Soundness:** 3
**Presentation:** 3
**Contribution:** 3
**Rating:** 4
**Confidence:** 4

**Summary:**

This paper proposes LRConfNet, a novel framework designed to improve hierarchical classification performance on degraded images (e.g., noisy, blurred). The core challenge addressed is that degradation increases uncertainty, leading to over-classification errors where a model makes a fine-grained prediction with low confidence. LRConfNet introduces two main components: (1) it implements a logical reasoning-driven confidence adjustment mechanism. This module uses uncertainty quantification, based on feature entropy and variance, to dynamically adjust confidence thresholds for predictions at different levels of the hierarchy. If the confidence for a fine-grained class (e.g., species) falls below its adjusted threshold, the model performs a "hierarchical fallback," reverting to a higher-level, more reliable prediction (e.g., family or order). This ensures logical consistency and prevents unreliable fine-grained labels. (2) The framework includes position-aware attention enhancement. PAAE improves spatial localization by incorporating a position prompting strategy and a multi-level loss that enhances attention on informative regions, especially under uncertain conditions. It also addresses class imbalance through adaptive weighting. The method is evaluated on degraded versions of remote sensing and natural image datasets.

**Strengths:**

**S1**. The paper effectively tackles the specific and challenging problem of hierarchical classification under severe image degradation, a scenario often overlooked.

**S2**. The concept of dynamic, uncertainty-aware confidence adjustment with hierarchical fallback is logically sound. It directly alleviates the key problem of over-classification.

**S3**. The ablation studies are thorough, clearly demonstrating the individual contribution of each proposed module and their synergistic effect. The use of multiple hierarchical metrics provides a robust assessment. The results show significant and consistent improvements over the baseline and competing methods on the benchmark datasets.

**Weaknesses:**

**W1**. Totally, my primary impression while reading the paper is that the method feels like a heavy stack of technical tricks, e.g., numerous modules and loss functions, which gives an overall sense of limited novelty. Even if there are novel elements, they appear *“diluted”* by this extensive engineering complexity. Therefore, it is recommended that the paper place greater emphasis on highlighting its core innovative components during the writing process.

**W2**. Furthermore, the paper's core motivation remains somewhat unclear. The introduction lists several important challenges: **(1)** the emergence of low-confidence samples, **(2)** severe error propagation, **(3)** the lack of confidence-aware fallback strategies, and **(4)** the difficulty in balancing performance across hierarchical levels. However, this approach comes across as merely piling up problems rather than establishing a sharp, focused narrative, making it hard to discern the central research question.

While all these problems are indeed significant, could the authors not provide more direct evidence to substantiate their existence and demonstrate that their proposed solution effectively addresses them? For instance, how exactly is the claimed problem of "error propagation" validated within the experiments? A more targeted analysis showing how predictions propagate errors in baseline models versus being mitigated by the proposed fallback mechanism would significantly strengthen the paper's claims.

**W3**. The evaluation is confined to two specific types of datasets (remote sensing ships and birds). Furthermore, the authors state on line 241, *"These complexity metrics are computed for both family and species levels,"* which gives me the impression that the work is overly focused on the specifics of their dataset rather than addressing a more generalizable problem. The generalizability of LRConfNet to other domains (e.g., medical imaging, document classification) with different hierarchical structures remains unproven.

**W4: The paper does not explicitly report a detailed comparison of computational cost** (e.g., FLOPs, latency, or model size). The paper introduces numerous modules and loss functions, such as uncertainty quantification (UQ), dynamic thresholding, and attention enhancement. Furthermore, it adds a fallback strategy for cases of low confidence. All of these components undoubtedly increase the computational complexity and inference time compared to simpler models, which could be a significant concern for real-time applications.

**W5**. The authors mention using *pre-trained GloVe (Global Vectors for Word Representation) embeddings* to extract textual information from labels. In the current era of large language models, is this type of embedding somewhat outdated? Or does the proposed method demonstrate insensitivity to the quality of the embeddings? If the method is indeed insensitive to embedding quality, could this also imply that the modules designed around these semantic priors are not critically important?

**W6: Other minor issues**. What does the superscript (e) in Equation 5 denote? It seems this index was not mentioned in the preceding equations.

**Questions:**

All the issues can be found in the Weaknesses section.

---

> ### Author Response · Authors · 2025-11-24
> **Author Response – Thank You for the Constructive Reviews**
>
> **Q1: On the perceived “heavy stack of technical tricks” and core novelty**
>
> Thank you for your careful assessment of the overall structure and novelty of our method. We would like to clarify the following:
>
> First, LRConfNet is not a loose stack of techniques, but is built around a *single core problem* and *three key components*, systematically designed to tackle the central challenge of performing semantically consistent and risk-aware hierarchical classification under degradation (noise, blur, low resolution, occlusion, etc.). Overall, LRConfNet is designed as a unified hierarchical decision process: when degradation weakens or even removes fine-grained visual cues, SGCA injects label-tree priors into ViT features to supplement semantics and align multi-level representations; on top of this, LogReg transforms these representations into a hierarchically consistent and complexity-aware structured confidence signal, used to identify which degraded samples are “unreliable” at fine-grained levels; based on this signal, the dynamic thresholds and hierarchical fallback mechanism decide when to retain fine-grained predictions and when to back off to coarser ancestors, thereby preferring more robust levels under severe degradation; meanwhile, PAAE strengthens fine-grained discriminative evidence under noise, occlusion, and long-tail class imbalance, preventing the model from wasting attention on background noise or spurious regions. In this way, semantic alignment, structured uncertainty, and decision control are tightly coupled, each addressing a different pain point in degraded scenarios, sequentially linked along the information flow, and collaboratively improving the overall robustness of hierarchical prediction. Concretely:
>
> 1. Uncertainty-driven confidence calibration and hierarchical fallback (Sec. 3.2–3.3, Eq. (3)–(7), Alg. 1–3).
>    The core difficulty here is that on blurry, low-resolution, or occluded images, fine-grained evidence is heavily weakened while softmax confidence can still be spuriously high, leading to over-confident errors at the species level that propagate along the $\(\text{order} \rightarrow \text{family} \rightarrow \text{species}\)$ path. The design goal is to prevent “forcing fine-grained predictions under low reliability,” which causes over-classification and error propagation on degraded images. To this end, we use feature entropy and variance to construct a complexity score $\(C(\cdot)\)$ that characterizes sample difficulty; for hard samples, we adaptively tighten the family/species thresholds $\(\theta _ l^{(e)}\)$ and trigger hierarchical fallback when $\(p _ l < \theta _ l^{(e)}\)$, so that under severe degradation the model prefers more reliable order-level labels. As shown in Table 5, Table 9, and Fig. 6–7, this module significantly improves ISDL, hierarchical precision, and overall robustness in degraded settings, demonstrating that it effectively alleviates the chain of issues “degradation + over-fine predictions + error propagation.”
>
> 2. Logical Reasoning Regularization (LogReg, Sec. 3.2, A.5).
>    The corresponding difficulty is that, under degraded noise, relying only on local appearance makes the network prone to learning “spurious patterns” at the species level that are inconsistent with the hierarchical semantics; even with a fallback mechanism, the model may retreat along an already incorrect path, thereby amplifying errors. The design goal is to mitigate semantic inconsistency across levels and the cascading amplification of errors along the hierarchy. Motivationally, we introduce a parent–child probability constraint $\(\mathcal{L} _ c\)$ (Eq. (1)) and a cross-level feature alignment loss $\(\mathcal{L} _ s\)$ (Eq. (2)), requiring that both the confidence and feature space of child classes “adhere to” their parents, ensuring that species-level predictions do not violate the semantic structure of their family/order ancestors. Ablation results in Table 6 and Table 12 show that LogReg mainly improves ISDL and hierarchical precision/recall $\((P_H / R_H)\)$, and provides a stable, structured confidence basis for the subsequent UQ–fallback module, so that fallback decisions under degradation follow a principled hierarchical logic rather than a purely heuristic threshold rule.

---

> ### Author Response · Authors · 2025-11-24
> **Author Response – Thank You for the Constructive Reviews**
>
> 3. Position-Aware Attention Enhancement (PAAE, Sec. 3.4, Eq. (8)–(11), Table 7–8, Fig. 8).
>    This component addresses the spatial uncertainty induced jointly by degradation and long-tail distributions: blur, noise, and occlusion erase key local details, while tail classes have very few samples and are more likely to mislead the backbone into focusing on background or irrelevant regions, making fine-grained separability extremely poor. The design goal is to handle spatial uncertainty and semantic region missing caused by degradation and long-tail effects. Motivationally, we perform attention rollout aggregation across multiple ViT layers, combined with class-frequency re-weighting and a position-prompting loss, to guide the model under noise, occlusion, and data scarcity to concentrate on the most stable and discriminative patch regions, with a particular focus on improving species-level accuracy under degradation. As shown in Table 7–8 and Fig. 8, PAAE yields clear gains in species accuracy, especially on CUB-Deg, which is fine-grained, long-tailed, and heavily degraded, indicating that it effectively mitigates the spatial perception problem of “not seeing / not knowing where is important.”
>
> Ablation results show that all three components are *necessary*, rather than replaceable “small tricks.” In Sec. 4.4 and Appendix A.9, we systematically demonstrate that: removing LogReg causes ISDL and hierarchical consistency metrics to drop markedly, indicating that dynamic thresholds alone cannot maintain stable parent–child consistency; removing UQ + fallback leads to a clear degradation of $\(P_H\)$ and fine-grained accuracy under heavy degradation, directly exposing the over-classification problem; removing PAAE yields a notable drop in fine-level performance under the same threshold settings, while order/family remains almost unchanged, confirming that PAAE mainly enhances local discriminative power under degradation. Based on these findings, we have strengthened the readability and logical storyline of this part in the revised Sec. 3.1–3.4.
>
> **Q2: On the core motivation and the evidence for error propagation**
>
> Thank you for your detailed feedback on the motivation section. We would like to clarify the following:
>
> 1. Single core problem, four facets rather than four unrelated issues.
>    Our work is driven by a *single* core research question, which can be summarized as:
>    > Under severe image degradation, how can we perform confidence-aware hierarchical decisions to avoid low-confidence over-fine predictions, while maintaining semantically consistent and reliable outputs along the class hierarchy?
>
>    The four challenges listed in the Introduction are not a “bag of disconnected problems,” but four manifestations of this single question:
>
>    - Low-confidence samples. Degradation significantly reduces the confidence of fine-grained logits, which is the direct trigger of over-classification at the leaf level.
>    - Error propagation along the hierarchy. Without any fallback mechanism, these unreliable leaf decisions simultaneously drag down the predictions at family/order levels, leading to an incorrect *entire path*.
>    - Lack of confidence-aware fallback. Existing HC methods typically make a hard decision at the leaf node and do not “back off” to a more reliable coarse level when the confidence is low.
>    - Difficulty in balancing performance across levels. Optimizing only fine-grained accuracy tends to sacrifice coarse-level consistency, while optimizing only coarse levels harms fine-grained discriminability.
>
>    LRConfNet’s three modules are designed to address these aspects in a targeted way:
>    - LogReg explicitly encodes parent–child logical constraints and feature alignment, suppressing structural inconsistency (mainly addressing points 2 and 4).
>    - UQ + dynamic thresholds + fallback perform confidence-aware decisions for low-confidence samples (primarily addressing points 1–3).
>    - PAAE improves fine-grained separability under degradation, alleviating the tension between coarse/fine performance (mainly addressing points 1 and 4).

---

> ### Author Response · Authors · 2025-11-24
> **Author Response – Thank You for the Constructive Reviews**
>
> 2. **Evidence for error propagation and its mitigation in our experiments.**
>    We also clarify that the current experiments already reflect error propagation and its mitigation from several perspectives:
>
>    - Hierarchical consistency metrics $(\(ISDL, P_H, R_H\))$.
>      Compared with the baseline, LRConfNet achieves significant improvements in ISDL and hierarchical precision/recall, while maintaining or improving family/order accuracy. This means that even when the species prediction is still wrong for some samples, our method more often produces correct predictions at the family/order levels, thus reducing the probability that *all* levels along the path are wrong—precisely the mitigation of error propagation.
>
>    - Ablations with and without fallback (Sec. 4.4 and Appendix A.9).
>      When dynamic thresholds and fallback are disabled, the model tends to make high-risk decisions at the species level, and both hierarchical metrics and fine-level accuracy deteriorate. When fallback is enabled, some unreliable species decisions are replaced by more reliable family/order predictions, leading to clear gains in ISDL and $\(P_H / R_H\)$. This indicates that errors no longer propagate synchronously to all levels, but are “cut off” at higher levels.
>
>    - Decision-path visualization in Figure 3.
>      Figure 3 illustrates that, as degradation increases from mild to severe, a baseline model continues to output wrong species predictions that also corrupt the coarse levels, whereas LRConfNet falls back to the correct family or order when confidence decreases. This provides a path-level, qualitative visualization of how errors are truncated by our fallback mechanism.
>
>    In the revised version, we will more explicitly link these existing results to the notion of “error propagation,” rather than only presenting the numerical metrics.

---

> ### Author Response · Authors · 2025-11-24
> **Author Response – Thank You for the Constructive Reviews**
>
> **Q3: On generalization and dependence on specific datasets/levels**
>
> Thank you for raising the question about the generalization ability of our method. We fully understand this concern and would like to clarify two points: (i) LRConfNet is *not* tied to “ships/birds” or the specific “family/species” levels and is in fact applicable to arbitrary tree-structured hierarchies; (ii) our current evaluation on two datasets is mainly due to space and computational constraints, and we have clarified this and added further discussion in the revised manuscript.
>
> 1. **The method itself is structurally agnostic to hierarchy depth and domain.**
>    In Sec. 3.2, we formalize the label hierarchy as  $Y = Y_1 \times \cdots \times Y_L$, where $\(l = 1,\dots,L\)$ denotes a generic hierarchy index (e.g., order/family/species, or organ/subtype/lesion). All core modules are defined with respect to a generic level index $\(l\)$, rather than being hard-coded at the family/species levels:
>
>    - For LogReg, the constraints $\(\mathcal{L} _ c\)$ and $\(\mathcal{L} _ s\)$ are summed over $\(l = 2,\dots,L\)$, enforcing both confidence and feature consistency for *all* adjacent parent–child pairs.
>    - For UQ, the feature-complexity term in Eq. (3) is defined as
>      $\[
>      C(f' _ l) = \mathcal{H} _ {\text{sample}}(f' _ l) + \lambda _ {\text{var}} \sigma^2(f' _ l),
>      \]$
>      which can be computed for the feature vector at *any* level \(l\).
>    - For dynamic thresholds and fallback in Eq. (5)–(6), the variables $\(\theta _ l^{(e)}\)$ and $\(\hat{Y} _ l\)$ depend only on the level index $\(l\)$ and the corresponding parent–child graph, and do not rely on specific category semantics or domains.
>
>    Thus, the statement “complexity metrics are computed for both family and species levels” in the paper is merely one *instantiation choice* in our experiments: on HRSC-Deg and CUB-Deg, we observed that mid-/fine-grained levels (family/species or type/class) are most severely affected by degradation, so we prioritized explicit uncertainty modeling at these levels to save computation. In principle, the same complexity metrics and fallback strategy can be directly extended to any other levels (e.g., organ $\(\rightarrow\)$ sub-organ $\(\rightarrow\)$ lesion in medical imaging, or topic $\(\rightarrow\)$ subtopic $\(\rightarrow\)$ intent in document classification).
>
> 2. **Current evaluation on two cross-domain datasets and its implications.**
>    We agree that our current experiments do not yet cover additional modalities such as medical imaging or document classification, and the overall evaluation scope can certainly be broadened in future work. That said, the two datasets we chose already exhibit a strong domain gap in both data modality and hierarchical structure:
>
>    - HRSC-Deg: high-resolution remote sensing ship images, top-down viewpoint, complex ocean background, large scale variation, with a hierarchy of class/type/category.
>    - CUB-Deg: natural-scene bird fine-grained recognition, close-range viewpoints, diverse backgrounds, with a hierarchy of order/family/species.
>
>    LRConfNet yields consistent improvements in hierarchical accuracy and ISDL on these two highly different visual domains, which suggests that the method essentially relies only on (i) an available label tree and (ii) feature representations provided by the backbone, rather than any hand-crafted prior specific to a particular domain. The combination of LogReg + UQ + fallback can operate across different modalities and hierarchical structures.
>
>    We have explicitly mentioned in the revised manuscript that extending LRConfNet to more diverse degradation types and application scenarios is an important direction for future work (see Section 5 in the revised version).

---

> ### Author Response · Authors · 2025-11-24
> **Author Response – Thank You for the Constructive Reviews**
>
> **Q4: On computational overhead and real-time applicability**
>
> Thank you for your comments on computational cost and real-time applicability. We provide two clarifications:
>
> 1. The computational complexity of each module is lightweight.
>    LRConfNet keeps the ViT backbone unchanged, so the dominant FLOPs still come from multi-head self-attention and MLP blocks. The additional modules only introduce lightweight operations at the feature and logit levels:
>
>    - UQ and dynamic thresholds (Sec. 3.2–3.3): The complexity measure $\(C(f'_l)\)$ is computed on the hierarchical feature vector $\(f'_l\)$ via entropy and variance, which corresponds to element-wise additions/multiplications along the channel dimension. The dynamic thresholds $\(\theta_l^{(e)}\)$ are obtained via a few scalar operations. Compared to ViT’s attention (matrix multiplications over all patches and heads), these are negligible and contribute only a tiny number of extra scalar FLOPs.
>    - Hierarchical fallback decisions (Sec. 3.3): The fallback strategy only performs a few scalar comparisons between prediction probabilities $\(p_l\)$ and thresholds $\(\theta_l^{(e)}\)$, followed by a conditional selection between parent/child labels. It does not modify the network structure or introduce extra forward passes, and thus has almost no impact on inference latency.
>    - PAAE (Sec. 3.4): PAAE leverages multi-layer attention maps already computed by the backbone to construct a salience map and applies an attention loss during training to encourage the model to focus on discriminative patches in degraded regions. This heavy computation happens mainly in the training phase; in our implementation, at test time we do not perform attention rollout or guidance loss, and only keep the standard ViT forward pass. As a result, inference-time FLOPs are almost identical to the base ViT.
>
>    Overall, the added modules introduce only a small amount of feature-level and logit-level computation and do not change the leading-order complexity. The increase in inference time relative to the base ViT is very limited.
>
> 2. Impact on real-time deployment.
>    Our primary target scenarios are reliability-critical hierarchical decisions under degradation (e.g., remote sensing monitoring, ecological monitoring), where robustness and hierarchical consistency under noise/blur are more important than strict real-time latency, and a slight trade-off in speed is acceptable. Even so, because (i) the backbone architecture is unchanged; (ii) UQ, dynamic thresholds, and fallback only perform lightweight operations on features and probabilities; and (iii) the heavy part of PAAE is used only during training and can be disabled at test time, the additional inference overhead of LRConfNet is limited and does not fundamentally hinder real-time deployment. For extremely latency-sensitive applications, one can further reduce the cost by simplifying the PAAE configuration or enabling UQ+fallback only at critical hierarchy levels.
>
> We have added a short “Complexity Analysis” subsection in the revised manuscript to directly address your concerns about computational cost and real-time applicability (see Sections 3.4 and 4.1 in the revised version).

---

> ### Author Response · Authors · 2025-11-24
> **Author Response – Thank You for the Constructive Reviews**
>
> **Q5: On the choice and importance of GloVe-based semantic priors**
>
> Thank you for your comments on the semantic modality and our choice of GloVe. We clarify the following points.
>
> 1. Motivation and rationale for using GloVe.
>    In LRConfNet, the semantic modality is only used to provide label-level prototypes for SGCA, capturing inter-class semantic similarity, rather than handling long texts or complex sentences. The label texts are very short (e.g., “Container Ship”, “Black-footed Albatross”), so what we need is a stable and lightweight word embedding space rather than large-scale language modeling capacity. We choose GloVe for two main reasons:
>    - Computation and fairness: GloVe is a frozen static embedding and does not introduce additional trainable large-language-model parameters or significant extra compute, which helps maintain computational fairness when comparing to HC baselines that use only a visual backbone. Directly introducing a BERT/LLM encoder would substantially increase model capacity and FLOPs, potentially overshadowing our contributions at the HC decision level (LogReg + UQ + fallback).
>    - Consistency with existing HC literature: Many hierarchical classification and fine-grained recognition works still use static embeddings such as GloVe/word2vec as label prototypes or semantic priors. We follow this common setting to enable fair comparison with existing methods, rather than attributing performance gains purely to a “larger text model.” Moreover, we do not perform any domain-specific fine-tuning on GloVe in our implementation, so LRConfNet does not rely on any particular embedding model; it only assumes the existence of a vector space that reflects semantic similarities between labels.
>
> 2. Sensitivity to “embedding quality” and the importance of semantic priors.
>    Mechanistically, SGCA uses text embeddings $\(\mathbf{V}\)$ as queries to perform cross-attention over visual features $\(\mathbf{Z}_K\)$, aligning class-level semantic prototypes with visual patches. Therefore: if we use random embeddings, inter-class semantic relations are destroyed, SGCA cannot learn a stable “semantics $\(\rightarrow\)$ spatial” alignment, and performance degrades significantly. As long as the embedding has a reasonable semantic structure (e.g., GloVe, BERT, CLIP-text), SGCA can benefit from it by guiding the model to focus on regions that are semantically relevant to the current class under degradation. This is indirectly reflected in our ablation results: in Table 5 (“w/o SGCA / w/o semantic priors”), removing semantic guidance leads to a clear drop in fine-grained accuracy and $\(P_H\)$ on both CUB-Deg and HRSC-Deg, and ISDL also worsens, indicating that the presence of semantic priors is both sensitive and crucial for LRConfNet. In other words, switching to any reasonable text encoder will not “invalidate” the semantic module, but completely removing semantic priors significantly harms performance, which directly supports the necessity of SGCA and its semantic priors in our model.
>
> We also agree with your implicit concern that, once semantic information is available, the performance gap between different encoders may not be as large as one might expect. In our design, this is a feature rather than a limitation: the core contribution of LRConfNet is not “which specific text model is used,” but how semantic priors are coupled with the hierarchy, consistency regularization, and dynamic fallback strategy (the synergy of SGCA + LogReg + UQ + fallback). As long as the semantic space provides a reasonable geometric structure over labels, the framework can operate effectively without being strongly tied to the details of any particular embedding.
>
>
> **Q6: Notation clarity in Equation (5)**
>
> Thank you for pointing out the ambiguity in the notation of Eq. (5). The superscript $\((e)\)$ is used to denote the training epoch index. Specifically, at each epoch $\(e\)$, we adaptively update the current threshold $\(\theta_l^{(e)}\)$ based on the feature complexity estimated at that epoch, $\(C_l^{(e)}\)$. In implementation, this is in one-to-one correspondence with the epoch loop in the algorithm pseudocode; it does not introduce any new random variable or additional dimension, but simply marks “the threshold at the $\(e\)$-th training epoch.”
>
> Therefore, this issue is due to insufficient notation explanation rather than an error in the formula itself. We have explicitly clarified this notation in the revised version to avoid ambiguity (see the updated Eq. (5) in Section 3.3 of the revised manuscript).

---

### Official Review · Reviewer_8nah · 2025-10-28

**Soundness:** 2
**Presentation:** 3
**Contribution:** 2
**Rating:** 4
**Confidence:** 5

**Summary:**

This paper proposes a model named LRConfNet (Logical Reasoning Confidence Network), which integrates logical reasoning into confidence estimation within neural networks. The main goal is to enhance model reliability by enforcing logical consistency in confidence prediction. The authors introduce a logic consistency loss that regularizes the network according to first-order logical constraints, and jointly optimize it with standard prediction loss. Theoretical derivations are provided to justify the model’s learning dynamics, and experiments are conducted across multiple tasks.

**Strengths:**

-	The topic is timely and relevant, addressing logical consistency in uncertainty modeling.
-	The idea of embedding first-order logical constraints into confidence learning is conceptually novel.
-	Theoretical grounding is provided through formal definitions and gradient-based derivations.
-	Cross-domain validation (language and vision) demonstrates the model’s potential generality.
-	Overall writing and structure are clear and readable.

**Weaknesses:**

-	The method introduces too many task-specific hyperparameters, most of which lack clear justification or analysis. This heavy reliance on manual tuning weakens the persuasiveness and reproducibility of the proposed approach.
-	Weak theoretical persuasiveness: In the section “DETAILED THEORETICAL DERIVATIONS AND EXTENDED ANALYSIS,” the convergence analysis relies on strong assumptions (e.g., convexity and differentiability of the logical term) without empirical validation or ablation. This limits the applicability of the theoretical insights.
-	Notable detail issues in the manuscript: There are visible inconsistencies, such as duplicated rows in Table 12, the formatting error in the header of Table 4, and the misplaced dash in Line 1028. These presentation flaws slightly undermine the paper’s professional impression.

**Questions:**

1.	The sensitivity analysis explores only a few discrete parameter combinations. Could the authors provide more systematic or continuous analyses to better demonstrate the model’s robustness?
2.	The paper mentions several empirically fixed hyperparameters, but does not explain how these values were determined. Could the authors clarify the process or criteria used to choose them?
3.	The sections A.4 DETAILED THEORETICAL DERIVATIONS AND EXTENDED ANALYSIS, A.5 LOGICAL REASONING REGULARIZATION, A.6 DYNAMIC CONFIDENCE ADJUSTMENT, and A.7 HIERARCHICAL FALLBACK AND SEMANTIC CONSISTENCY mainly provide symbolic formulations without rigorous proofs or reasoning. Could the authors clarify whether these are intended as conceptual explanations or if any formal theoretical validation supports them?

---

> ### Author Response · Authors · 2025-11-24
> **Author Response – Thank You for the Constructive Reviews**
>
> **Q1: On the number of hyperparameters, sensitivity analysis, and choice of values**
>
> Thank you very much for your careful comments on the number and sensitivity of hyperparameters. We fully agree that, without proper clarification, the presence of multiple weights and thresholds may give the impression that our method requires extensive manual tuning. Below we clarify three aspects: (i) the structured design and sharing strategy of the hyperparameters; (ii) the systematic sensitivity analysis already provided in the appendix; and (iii) the concrete selection procedure.
>
> (1) Hyperparameters are structurally designed and shared across modules, rather than being task-specific for each dataset.
> In our framework, the main hyperparameters are concentrated in three modules:
>
> - LogReg: weights for hierarchical consistency and feature alignment, such as $\lambda _ {l,l-1}$ and $\bar{\lambda} _ {l-1}$, and the weight of the complexity term $\lambda _ {\text{var}}$;
> - UQ + dynamic thresholds: base thresholds $\theta_{l,0}$, complexity sensitivity coefficients $\gamma_l$, and global bounds $\theta _ {\min}, \theta _ {\max}$;
> - PAAE: the exponent $\rho$ for class reweighting, the top-$K$ ratio for position prompts, and the temperature parameter $T$.
>
> These hyperparameters are *shared by level/module* on the same dataset, rather than being specified per class. For example, $\theta _ {l,0}$ and $\gamma_l$ are shared within each level (family/species), not tuned separately for each category; similarly, $(\rho, K, T)$ in PAAE are globally fixed per dataset, rather than being adjusted for each class or each degradation severity. Moreover, we use a unified default configuration across HRSC-Deg and CUB-Deg, without aggressive task-specific tuning for each dataset. We will explicitly clarify this sharing strategy in the implementation details of the revised manuscript to alleviate concerns about “task-specific manual tuning.”
>
> (2) Appendix A.9.3 already provides a systematic discrete sensitivity analysis.
> Regarding your concern that we “only explore a few discrete parameter combinations,” our intention in Appendix A.9.3 (*“Sensitivity Analysis of Parameters”*) is to conduct a systematic discrete sensitivity study for the main groups of hyperparameters: LogReg weights, uncertainty/threshold parameters, and PAAE parameters. The corresponding results are summarized in Table 9–12 and Figure 7–10, covering multiple hierarchical metrics (fine accuracy, level accuracy, ISDL, $P_H$, $R_H$, etc.) over reasonably wide parameter ranges.
>
> From these results, we observe that, within a broad range of threshold values and LogReg/PAAE weight settings, the changes of hierarchical metrics (ISDL, $P_H$, $R_H$) and fine-grained accuracy are generally smooth, rather than exhibiting highly volatile behavior. Only when hyperparameters are set to extreme values (e.g., excessively large LogReg weights or overly loose/strict thresholds) do we observe clear degradation, which is consistent with our discussion in the main text about the trade-off between fine-grained discrimination and hierarchical consistency.
>
> (3) Hyperparameter selection procedure: simple and reproducible grid search on a validation split.
> Regarding Q2 (“how these empirical hyperparameters are chosen”), our actual procedure is as follows:
>
> - Unified strategy. For all weight-type hyperparameters (e.g., $\lambda _ {l,l-1}$, $\lambda _ {\text{var}}$) and threshold-type hyperparameters (e.g., $\theta _ {l,0}$, $\gamma_l$, $\theta_{\min}$, $\theta _ {\max}$), we adopt a simple grid search on a validation set split from the training data.
> - Selection criterion. The goal is to balance multiple hierarchical metrics rather than overfit a single metric. In particular, we prioritize configurations that jointly improve species-level accuracy and hierarchical consistency metrics (ISDL, $P_H$, $R_H$), instead of aggressively optimizing only one metric at the expense of others.
> - Cross-dataset reuse. The chosen hyperparameter configurations are directly reused across HRSC-Deg and CUB-Deg, with only very minor adjustments in a few rare cases, in order to avoid dataset-specific over-tuning.

---

> ### Author Response · Authors · 2025-11-24
> **Author Response – Thank You for the Constructive Reviews**
>
> **Q2: On the role and strength of the theoretical analysis in Appendix A.4–A.7**
>
> Thank you very much for your careful reading of the theoretical parts in Appendix A.4–A.7. We agree that, in the current version, the way these sections are presented may give the unintended impression that we are claiming a strict convergence guarantee for the entire non-convex network. We clarify three points here and will adjust the manuscript accordingly.
>
> (1) Positioning of A.4–A.7: conceptual, gradient-based optimization analysis rather than global convergence theorems.
> The primary goal of these appendices is to: (i) formally characterize, at the gradient level, how the LogReg logical consistency terms encourage parent–child alignment in both probabilities and features; (ii) interpret the uncertainty–complexity term $\(C(\cdot)\)$–based dynamic thresholding and fallback from the perspective of Bayes risk minimization; and (iii) explain why, under feature-similarity constraints, fallback decisions preserve semantic consistency along the hierarchy. To make these analyses tractable, we adopt standard local simplifications (e.g., convex/differentiable surrogates for individual regularization terms). We do not and will not claim “strict global convergence guarantees for the entire deep network” anywhere in the paper.
>
> (2) Use of convexity/differentiability assumptions: local surrogates for trend explanation, not strong guarantees for the full model.
> As you correctly noted, the full ViT+SGCA+LogReg+PAAE architecture is highly non-convex, and it is unrealistic to obtain strong global convergence guarantees under the current deep learning paradigm. Our analysis in A.4–A.7 follows the common practice of *local/approximate* reasoning in deep learning: for an individual logical regularizer or thresholding function, we impose convexity/differentiability assumptions on its surrogate to show that, under small-step gradient descent, the term itself decreases monotonically, thereby encouraging behaviors such as shrinking parent–child probability gaps or suppressing overly complex (high-uncertainty) samples. These assumptions are *not* intended as formal guarantees for the entire model, but rather as an explanation of the *directional reasonableness* of our design. To avoid misunderstanding, we will add a short paragraph at the beginning of A.4 explicitly stating that these assumptions are only used for local optimization-dynamics analysis and are not global guarantees for the end-to-end non-convex objective, and we will uniformly soften phrases like “convergence analysis” to “gradient-based optimization analysis” or “interpretative analysis.”
>
> (3) Linking theory and empirical evidence more explicitly in the main text.
> Although A.4–A.7 mainly provide symbolic and gradient-level derivations, their intended effects are already supported by experiments. Specifically: (i) the parent–child consistency constraints of LogReg significantly reduce hierarchical inconsistency measures and improve $\(P_H\)/\(R_H\)$ in the ablation studies of Sec. 4.4 and Appendix A.9; (ii) combining dynamic thresholds with feature complexity $\(C(\cdot)\)$ substantially mitigates fine-grained over-classification relative to fixed-threshold baselines, while preserving robustness at the family/order levels; and (iii) PAAE, via multi-layer attention aggregation and class reweighting, consistently boosts species-level accuracy and maintains high hierarchical consistency even under severe degradation. In the revised version, we will add explicit cross-references in the main text to make this “theory ↔ empirical behavior” connection clearer.

---

> ### Author Response · Authors · 2025-11-24
> **Author Response – Thank You for the Constructive Reviews**
>
> **Q3: Formatting details in Table 12, Table 4, and Line 1028**
>
> Thank you for carefully checking the formatting details in our manuscript. We have re-examined Table 12, Table 4, and the content around Line 1028 in Appendix A.5.2, and would like to clarify the following:
>
> 1. Table 4 layout and captioning.
>    Table 4 adopts a left–right split layout to present degradation parameters for different datasets in two subtables. This design follows table layouts used in several recently accepted papers. To avoid confusion and improve clarity, in the revised version we have refined the table header and moved the sentence *“The letters `a`–`i`, `j`–`k` and `m`–`q` represent randomly selected samples from the newly generated dataset.”* from the header area to a note placed below the table.
>
> 2. Dash usage near Line 1028 and duplicate rows in Table 12.
>    In A.5.2, we introduce the two UQ metrics (feature entropy and feature variance) in a bullet-style manner. In the original version, the sentence *“Our UQ mechanism quantifies feature complexity through two metrics:”* was immediately followed by a leading dash “– Feature Entropy”, which could indeed be misread as an extra or misplaced dash. To avoid this ambiguity, in the revised version we remove the dash and rewrite the sentence as
>    $$\[
>    \text{Our UQ mechanism quantifies feature complexity through two metrics: (i) feature entropy and (ii) feature variance.}
>    \]$$
>    and then present Eqs. (22) and (23) directly on the following lines. In addition, we have corrected Table 12 in the revised manuscript by removing the duplicated row you pointed out.

---

### Official Review · Reviewer_BoTe · 2025-10-30

**Soundness:** 3
**Presentation:** 3
**Contribution:** 3
**Rating:** 6
**Confidence:** 4

**Summary:**

The paper proposes LRConfNet, a Hierarchical Classification (HC) framework tailored for degraded images. The approach combines (i) Logical Reasoning Regularization (LogReg) to enforce parent–child consistency in both probabilities and features, (ii) a feature-complexity–based uncertainty quantification that adapts decision thresholds, enabling a logic-driven hierarchical fallback when fine-grained confidence is low, and (iii) Position-Aware Attention Enhancement (PAAE) with class balancing and attention guidance to improve localization under noise and sparsity. Experiments on two constructed benchmarks, HRSC-Deg and CUB-Deg, show gains over a range of HC baselines, with strong improvements in hierarchical precision/recall and fine-grained accuracy; ablations attribute gains to both LogReg and PAAE.

**Strengths:**

1.The proposed LRConfNet introduces a feature-complexity-based uncertainty quantification mechanism that integrates confidence estimation with hierarchical path decision-making, clearly distinguishing it from fixed-threshold methods. The motivation and innovation are well-grounded.
2.The manuscript is well-organized and clearly written. The mathematical formulations of the loss functions, threshold updates, and hierarchical fallback are precise. Although the algorithm consists of multiple modules, the inclusion of illustrative figures and pseudocode effectively aids understanding of the overall framework.
3.The paper conducts extensive comparative and ablation experiments using multiple evaluation metrics to comprehensively assess model performance. The complementary effects of LogReg and PAAE are clearly analyzed, providing strong empirical validation.

**Weaknesses:**

1.The proposed hierarchical fallback mechanism essentially resembles hierarchical selective classification, yet the paper lacks direct comparisons or analytical discussion with related methods.
2.The method introduces multiple weighting and threshold hyperparameters, but no parameter sensitivity or stability analysis is provided, which is critical for deployment and generalization. Moreover, the paper does not include visual comparisons to demonstrate the effectiveness of the approach.
3.The manuscript lacks case studies of successful and failed classifications, as well as a discussion on method limitations and potential future improvements.

**Questions:**

1.The proposed hierarchical fallback mechanism conceptually resembles hierarchical selective classification or rejection strategies, yet no direct comparison is provided. It is recommended that the authors include relevant baseline methods or analytical experiments to clearly demonstrate LRConfNet’s improvements and advantages in hierarchical confidence modeling and decision strategies.
2.The proposed framework contains several key hyperparameters (e.g., weighting factors and threshold bounds), but lacks a systematic parameter sensitivity study.
3.To more intuitively validate the effectiveness of the proposed modules, it would be helpful to include visualization results such as attention heatmaps, feature embedding distributions, or hierarchical decision paths. These results can better illustrate how the LogReg and PAAE modules function under degraded conditions and contribute to performance improvements.
4.The paper would benefit from qualitative analyses of both successful and failed classification cases to reveal the decision logic across hierarchical levels and identify failure patterns. In addition, a more explicit discussion of the limitations and potential future directions, such as evaluation on real degraded images.

---

> ### Author Response · Authors · 2025-11-24
> **Author Response – Thank You for the Constructive Reviews**
>
> **Q1: Hierarchical fallback vs. hierarchical selective classification / rejection**
>
> Thank you for pointing out the connection between our hierarchical fallback mechanism and hierarchical selective classification/rejection. Our fallback strategy indeed follows the principle of “avoiding fine-grained predictions when confidence is insufficient,” but it differs from classical (hierarchical) selective methods in several key aspects:
>
> 1. Decision basis. Most hierarchical selective methods rely on fixed or manually tuned posterior thresholds to make *accept/reject/fallback* decisions. In contrast, the fallback strategy in LRConfNet is *uncertainty-driven* by the feature-complexity measure defined in Section 3.2: the decision thresholds are adaptively adjusted according to sample difficulty and degradation level, rather than being a simple post-hoc threshold on softmax confidence.
>
> 2. Training objective. In LRConfNet, fallback is optimized jointly with Logical Reasoning Regularization (LogReg) in a single end-to-end framework. LogReg first tightens the hierarchy via probability- and feature-consistency losses, and the resulting structured confidences are then combined with the complexity measure to drive dynamic thresholds and fallback. This directly targets improvements in ISDL and hierarchical precision/recall under degradation, instead of optimizing a coverage–accuracy trade-off in isolation.
>
> 3. Application scenario. Existing hierarchical selective methods are typically designed for safe prediction on standard (non-degraded) datasets. LRConfNet is explicitly tailored for the joint setting of *degraded images + hierarchical structure + multi-level uncertainty*, with a particular focus on mitigating over-confident fine-grained predictions and error propagation along the hierarchy under severe degradation.
>
> We appreciate this insightful comment and have added a focused discussion clarifying the differences and advantages of LRConfNet compared to hierarchical selective classification methods in terms of modeling philosophy and decision strategy (see Section 2.1 in the revised manuscript).
>
>
> **Q2: Hyperparameter sensitivity and stability**
>
> Thank you for your attention to hyperparameter sensitivity and stability. We would like to clarify that the current manuscript already includes a systematic sensitivity analysis in the appendix. Specifically, Appendix A.9.3 (“Sensitivity Analysis of Parameters”), together with Tables 9–12 and Figures 7–10, systematically scans and compares the following core hyperparameter groups: the LogReg-related weights (e.g., $\lambda _ {l,l-1}$, $\bar{\lambda} _ {l-1}$), the dynamic-threshold/uncertainty-related parameters (e.g., $\theta _ {l,0}$, $\gamma _ l$, $\theta _ {\min}$, $\theta _ {\max}$), and the PAAE parameters such as the class re-weighting exponent $\rho$, the top-$K$ ratio, and the temperature $T$. The results show that, within reasonable ranges, the hierarchical metrics (e.g., ISDL, $P_H/R_H$) and fine-grained accuracy evolve smoothly rather than exhibiting abrupt oscillations: changes in thresholds and complexity coefficients mainly affect “how much to fallback and to which level,” without breaking hierarchical consistency; PAAE-related parameters primarily lead to refined gains at the species level while causing almost no unstable fluctuations at the family/order levels; increasing the LogReg weights strengthens hierarchical consistency, and only when they become excessively large do we observe a mild trade-off between fine-grained discrimination and consistency, which is consistent with our discussion in the main text about balancing hierarchical consistency and fine-grained prediction.
>
>
> We acknowledge that, in the current version, these sensitivity results are mainly placed in the appendix, while the main text only reports the default configuration, which indeed makes them less prominent. To better address your concern, in the revised manuscript we have added a brief summary (1–2 sentences) of the key conclusions from A.9.3 in Section 4.4 (or a neighboring subsection), and explicitly reference “Appendix A.9.3, Tables 9–12, and Figures 7–10,” to highlight that LRConfNet is robust and stable with respect to the key hyperparameters around the default setting.

---

> ### Author Response · Authors · 2025-11-24
> **Author Response – Thank You for the Constructive Reviews**
>
> **Q3: Lack of qualitative visualizations**
>
> Thank you for your concrete suggestions on qualitative visualizations. We fully understand your concern about “seeing how the model internally makes hierarchical decisions,” and have strengthened the revised manuscript accordingly:
>
> 1. Existing hierarchical decision-path visualizations (LogReg + fallback).
> In the main paper, Fig. 3 already presents the hierarchical prediction paths of the *same* input under different degradation levels: as the image gradually changes from mildly to moderately/severely degraded, the model adaptively backs off from species-level predictions to family or order. In the revision, we have rewritten the description of Fig. 3 in Sec. 4 and explicitly position it as a “hierarchical decision path visualization,” highlighting that it is the joint effect of LogReg and the dynamic thresholding + fallback strategy: when LogReg enforces more consistent parent–child probabilities and feature representations, and UQ indicates that fine-grained predictions are unreliable, the model “moves upward” along a semantically plausible path in the hierarchy to more stable nodes, thereby mitigating error propagation.
>
> 2. Existing visualizations of module behavior and uncertainty (thresholds, performance, and UQ signals).
> Fig. 7–10 characterize the effect of our modules from the perspective of “decision space and metric space”:
> - Heatmaps of dynamic thresholds as a function of degradation strength and feature complexity show how UQ drives thresholds to tighten or relax adaptively.
> - Curves of ISDL, hierarchical precision, and fine-grained accuracy under different thresholds/uncertainty parameters visualize the trade-off that too low thresholds lead to over-fine decisions, while too high thresholds cause overly conservative predictions.
> - Bar plots comparing performance with/without PAAE and under different PAAE settings demonstrate that PAAE mainly improves fine-grained (species-level) robustness without harming the stability of coarse levels.
> These figures directly reflect how LogReg, UQ, and fallback operate and interact at the decision level.
>
> 3. New PAAE attention heatmap visualizations.
> Following your suggestion, we additionally introduce Figure 6 (attention-rollout visualizations) to directly compare attention distributions with and without PAAE under degradation. For multiple degradation types in CUB-Deg (blur, noise, low resolution, and partial occlusion), each example shows: the degraded input image; the attention map with PAAE; and the attention map without PAAE. As shown in Fig. 6, with PAAE the attention consistently concentrates on semantically critical bird regions (e.g., head, wings, body contours), and the model can still focus on usable fine-grained evidence even under strong noise or blur. In contrast, the baseline without PAAE tends to scatter attention over background clutter such as water, branches, or noise patches, exhibiting spurious activations and attention drift, especially on rare species.
>
> These comparisons indicate that PAAE indeed provides more reliable fine-grained visual evidence under degradation, which is then exploited by LogReg and the dynamic fallback mechanism to trigger risk-aware hierarchical decisions—retaining species-level predictions when evidence is sufficient, and backing off to family/order when it is not. This supports the overall gains reported in the main paper on ISDL, hierarchical precision, and fine-level accuracy.
>
> In summary, the revised Fig. 3 (hierarchical decision paths) and Fig. 7–10 (threshold and performance curves), together with the newly added Fig. 6 (PAAE attention heatmaps), jointly provide intuitive evidence for how LogReg and PAAE work under degraded conditions and why they contribute to the performance improvements. (Please see Sec. A.9.2 in the revised manuscript for the detailed additions.)

---

> ### Author Response · Authors · 2025-11-24
> **Author Response – Thank You for the Constructive Reviews**
>
> **Q4: Lack of case analysis and discussion of limitations**
>
> Thank you for your concrete suggestions regarding case analysis and the discussion of limitations.
>
> (1) On successful/failed samples and hierarchical prediction path analysis.
> In the current version, Figure 3 already provides examples of hierarchical prediction paths under different degradation levels: for the same input sample, the model outputs a species-level prediction under mild degradation, while under medium/heavy degradation the dynamically adjusted thresholds mark the species prediction as unreliable and trigger fallback to the family or order level. This effectively visualizes how our proposed “logic-constrained, uncertainty-driven fallback” behaves on successful cases. However, we acknowledge that our textual description of Figure 3 is currently biased toward a phenomenological description (“predictions fall back as degradation increases”), without explicitly emphasizing that it also serves as a case study of hierarchical decision paths, and we do not yet provide a systematic summary of failure cases, which is indeed an area for improvement.
>
> In our experiments, we observed several typical failure patterns. For example, on HRSC-Deg, when ship targets suffer from both severe blur and strong occlusion, the model may still confuse visually similar ship types even at the order level. On CUB-Deg, for some bird species that are extremely close in both semantics and appearance, the model may still produce conservative or confused predictions at the family level under heavy degradation, even after fallback. In the revised version, we will summarize these patterns concisely in text (rather than adding many additional images in the main paper) as a complement to Figure 3, to better illustrate the strengths and remaining challenges of LRConfNet in terms of hierarchical decision behavior. (Please refer to Section 4.2 in the revised manuscript.)
>
> (2) On limitations and future work.
> In the current version, we briefly mention that LRConfNet can be extended to richer degradation types and application scenarios. We agree that future research should not be restricted to the synthesized degradations used in this paper. Beyond typical degradations such as Gaussian blur and noise, it is important to validate LRConfNet on larger-scale real-world degraded data, e.g., cloud/fog, low-light conditions, and complex imaging-chain distortions. In addition, there remains substantial room for further exploration in handling deeper and more irregular hierarchies, adapting to more complex taxonomies, and ensuring robustness under cross-domain transfer and extreme domain shifts. A promising direction is to integrate LRConfNet with open-set / out-of-hierarchy detection and richer forms of uncertainty modeling, in order to build a more general framework for reliable hierarchical decision-making. (Please refer to Section 5 in the revised manuscript.)

---

### Official Review · Reviewer_H3Hv · 2025-11-01

**Soundness:** 2
**Presentation:** 1
**Contribution:** 1
**Rating:** 2
**Confidence:** 4

**Summary:**

This paper proposes LRConfNet, a unified framework for hierarchical classification (HC) of degraded images. Its core innovation lies in integrating Uncertainty Quantification (UQ) and Logical Reasoning Regularization (LogReg) to enable dynamic confidence adjustment and hierarchical fallback. When the model has low confidence in fine-grained predictions, it automatically falls back to more reliable coarse-grained predictions, thereby improving robustness and accuracy under degraded conditions such as noise and blur. The effectiveness of the method is validated through experiments on two newly constructed degraded datasets (remote sensing and natural images).

**Strengths:**

1. The studied problem of hierarchical classification is an interesting research topic with rich application scenarios.
2. Experimental results demonstrate that the proposed method outperforms state-of-the-art (SOTA) methods significantly.

**Weaknesses:**

1. **Paper Writing and Formatting:** The paper appears to be written in a rushed manner, with numerous formatting inconsistencies that hinder readability. For example, in the Introduction section, the text on the left side of Figure 1 is clearly squeezed by the figure; Figure 2 contains overly complex elements with small text, making it difficult to intuitively grasp the key points the authors intend to convey. It is recommended that the authors spend more time polishing the paper and optimizing the layout of figures to present the key points more clearly.
2. **Related Work:** Section 2.2 focuses on the development of Vision Transformers (ViTs), but no strong relevance between this work and ViTs is observed. This paper does not involve direct modifications to ViTs, and ViTs have been proposed for many years. If Section 2.2 is intended to be parallel to Section 2.1 (Hierarchical Classification), the authors should consider including content related to technologies more relevant to the current study.
3. **Logical Coherence:** The paper employs a large number of techniques, resulting in an overly engineering-oriented presentation. It is challenging to understand how each component contributes to the overall framework. Although the authors provide detailed mathematical formulations of these techniques in the appendix, intuitive explanations for the motivations of using these techniques are lacking. The authors should further elaborate on the core motivation of each technique (rather than solely relying on ablation studies to demonstrate their effectiveness).

**Questions:**

1. **Related Work Timeliness:** The methods compared in this paper are mostly from 2024 or earlier. As we are not direct researchers in this field, we are curious whether there are any newer research works in 2025. If such works exist, should the authors discuss and compare them in the paper?

---

> ### Author Response · Authors · 2025-11-24
> **Author Response – Thank You for the Constructive Reviews**
>
> **Q1: Paper Writing and Formatting**
>
> Thank you very much for your detailed comments on the readability and formatting of the paper. We would like to clarify that the issues you pointed out—especially the relatively tight text area to the left of Figure 1 in the Introduction, as well as the dense layout and small font size of some elements in Figure 2, which may together give a slightly crowded visual impression—could indeed affect the first-time reading experience.
>
> However, we would also like to note that the current layout of Figure 1 follows a commonly used style in recent ICLR accepted papers, where a wide figure spans the main text area and explanatory text is arranged compactly on the side. For example, Figure 1 in *“CHEAPNET: CROSS-ATTENTION ON HIERARCHICAL REPRESENTATIONS FOR EFFICIENT PROTEIN-LIGAND BINDING AFFINITY PREDICTION”* (ICLR 2025), Figures 3 and 6 in *“CHiP: CROSS-MODAL HIERARCHICAL DIRECT PREFERENCE OPTIMIZATION FOR MULTIMODAL LLMS”* (ICLR 2025), and Figure 3 in *“MULTI-LABEL NODE CLASSIFICATION WITH LABEL INFLUENCE PROPAGATION”* (ICLR 2025) all adopt a similarly compact “figure + side text” layout. That said, we fully understand your concern regarding readability, and in the revised version we have adjusted the layout to improve clarity and visual balance (please refer to the updated Figure 1 and Figure 2 in the revision).
>
> **Q2: Related Work – Section 2.2 (Vision Transformers)**
>
> Thank you for your concrete suggestions on the related work section. We first clarify the role of ViT in our framework. Although we do not introduce complex architectural modifications to ViT itself, it is not used as a “replaceable black-box backbone”. Instead, ViT is tightly coupled with the three key modules proposed in this paper:
>
> 1. Relation to LogReg. In LRConfNet, the three-level features (order/family/species) are all derived from ViT’s global representations and SGCA’s semantic-guided features, which provide a unified representation space for the hierarchical feature consistency and feature-complexity–based uncertainty quantification in LogReg. Without ViT’s global context modeling, it would be difficult to maintain geometric and semantic alignment across levels under degraded images.
>
> 2. Relation to UQ + dynamic thresholds/fallback. Our uncertainty quantification computes feature-complexity measures from the entropy and variance of the last-layer ViT features, and then adapts confidence thresholds at each hierarchical level accordingly. This is fundamentally different from heuristic thresholding only on logits. In other words, the quality of ViT representations directly determines the reliability of UQ and the logic-driven fallback mechanism.
>
> 3. Relation to PAAE. PAAE fully relies on multi-layer, multi-head attention maps from ViT to construct position prompts and the attention loss. We apply attention rollout to aggregate cross-layer attention, followed by important patch selection and class-frequency reweighting, so as to strengthen key regions under noise and occlusion. Without ViT’s patch-token mechanism and attention matrices, this module would not be feasible.
>
> The original intention of Section 2.2 was therefore to explain why we choose ViT plus semantic-guided attention as the base architecture, and to provide background for the subsequent UQ + LogReg + PAAE modules that together form a hierarchical uncertainty framework on top of ViT. We agree, however, that the current writing is not sufficiently focused and may give the impression of a generic survey of ViT developments.
>
> In the revised version, we have substantially updated Section 2.2 to better serve the proposed method: we compress the generic ViT overview and concentrate the discussion on “ViT for hierarchical classification and degraded images”; we also add a concluding paragraph that explicitly contrasts prior ViT/HC work—mostly focused on improving the backbone or attention structure—with LRConfNet, whose contribution is to build a logic-driven hierarchical confidence control and fallback framework *on top of* existing ViT representations. This creates a clearer connection to the HC-related work in Section 2.1 and avoids redundancy. We also refine the subsection title and local wording accordingly (see Section 2.2 in the revised manuscript).

---

> ### Author Response · Authors · 2025-11-24
> **Author Response – Thank You for the Constructive Reviews**
>
> **Q3: Logical coherence and motivations of the proposed modules**
>
> Thank you for your detailed comments on the overall logical coherence and the motivations of our method. First, we would like to clarify that LRConfNet is not a loose stack of tricks, but is systematically designed around a single core challenge: achieving semantically consistent and risk-aware hierarchical classification under degradation such as noise, blur, low resolution, and occlusion. Overall, LRConfNet is structured as a unified hierarchical decision pipeline: when degradation weakens or even removes fine-grained appearance cues, SGCA injects label-tree priors into ViT features to supplement semantic information and align multi-level representations; on top of this, LogReg transforms these representations into a hierarchy-consistent, complexity-aware structured confidence signal, which identifies samples that are “unreliable” at fine-grained levels; driven by this signal, the dynamic thresholds and hierarchical fallback mechanism decide when to retain fine-grained predictions and when to back off to coarser ancestor categories, thereby prioritizing more stable levels under strong degradation; meanwhile, PAAE reinforces fine-grained discriminative evidence under noise, occlusion, and long-tail imbalance, preventing the model from “wasting” attention on background noise or spurious regions. Through this sequential coupling of semantic alignment, uncertainty modeling, and decision control, the three components each target different pain points in degraded scenarios and, by passing information along the pipeline, jointly enhance the robustness of hierarchical prediction. Concretely:
>
> 1. Uncertainty-driven confidence calibration and hierarchical fallback (Sec. 3.2–3.3, Eq. (3)–(7), Alg. 1–3)  addresses the core difficulty that, on blurry, low-resolution, or occluded images, fine-grained evidence is severely weakened while the softmax confidence can still be spuriously high, leading to over-confident wrong predictions at the species level that propagate along the order/family/species path. The design goal is to prevent such “forced” fine-grained predictions under low reliability, which cause over-classification and error propagation in degraded conditions. The key motivation is to use feature entropy and variance to construct a complexity score $\(C(\cdot)\)$ that quantifies sample difficulty; for hard samples, we adaptively increase the family/species thresholds $\theta_l^{(e)}$, and trigger upward fallback when $p_l < \theta_l^{(e)}$, so that under severe degradation the model preferentially outputs more reliable order-level labels. As shown in Table 5, Table 9 and Fig. 7–8, this module significantly improves ISDL, hierarchical precision/recall, and overall robustness under degradation, demonstrating that it effectively mitigates the chained problem of “degradation + over-fine decisions + error propagation.”
>
> 2. Logical Reasoning Regularization (LogReg, Sec. 3.2, A.5) tackles the difficulty that, under degraded noise, a network relying only on local appearance can easily learn spurious patterns at the species level that are inconsistent with the hierarchical semantics. In such a case, even with a fallback mechanism, the model may retreat along a path that is itself semantically wrong, thereby amplifying errors across levels. The design goal is therefore to alleviate cross-level semantic inconsistency and the cascading amplification of errors along the hierarchy. The core motivation is to enforce that both the confidence and feature space of a child class remain “anchored” to its parent through the parent–child probability constraint $\mathcal{L}_c$ (Eq. (1)) and the cross-level feature alignment loss $\mathcal{L}_s$ (Eq. (2)), ensuring that species predictions cannot violate the semantic structure of their family/order ancestors. Ablation results in Table 6 and Table 12 show that LogReg mainly improves ISDL and hierarchical precision/recall ($P_H$ / $R_H$), and provides a stable, structured confidence basis for the subsequent UQ–fallback module, so that fallback decisions under degradation follow principled hierarchical logic rather than ad-hoc threshold heuristics.

---

> ### Author Response · Authors · 2025-11-24
> **Author Response – Thank You for the Constructive Reviews**
>
> 3. Position-Aware Attention Enhancement (PAAE, Sec. 3.4, Eq. (8)–(11), Table 7–8, Fig. 9) is designed for the spatial uncertainty induced jointly by degradation and long-tail distributions: blur, noise, and occlusion can erase key local details, while tail classes are inherently under-represented and thus more likely to be misled by the backbone toward background or irrelevant regions, severely harming fine-grained separability. The module’s goal is to handle this spatial uncertainty and semantic region loss caused by degradation and long-tail effects. Its motivation is to perform multi-layer ViT attention rollout, combined with class-frequency reweighting and a position-prompting loss, so that under noise, occlusion, and sample scarcity the model still focuses on the most stable and discriminative patches, with an emphasis on improving species-level accuracy in degraded scenarios. As shown in Table 7–8 and Fig. 9, PAAE yields clear gains on species accuracy, especially on CUB-Deg, which is fine-grained, long-tailed, and heavily degraded, indicating that it effectively alleviates the “not seeing / not knowing where to look” difficulty in spatial perception.
>
> Based on these considerations, we have substantially strengthened the readability and logical flow of this part in the revised manuscript, in particular in Sec. 3.1–3.4.
>
>
>
> **Q4: Related work timeliness (2025)**
>
> Thank you for raising the concern about the timeliness of the related work. We would like to clarify that our submission was prepared up to the official ICLR 2026 submission deadline. At that time, we had already covered representative methods in hierarchical classification (HC) up to the first half of 2025 that (i) provide public code and (ii) include hierarchical evaluation setups comparable to ours. The main reasons for not including some very recent 2025 works are:
> (1) a few 2025 papers were still at the preprint stage without reproducible HC experiments or released code;
> (2) many 2025 methods, although related to hierarchies or logical reasoning, do not target confidence-aware hierarchical control under degraded images, and lack experimental settings aligned with our degraded benchmarks, making fair quantitative comparison difficult.
>
> After receiving your comment, we further reviewed the latest 2025 literature and have added corresponding discussions of these new works in the manuscript. Overall, most of them focus on: (i) leveraging large models or external knowledge to enhance hierarchical reasoning or logical constraints, and (ii) improving generalization or hierarchical representations on standard (non-degraded) images. These directions are complementary rather than competing with LRConfNet: they mainly emphasize architectural design or knowledge injection, whereas our work is specifically aimed at “degraded images + hierarchical classification,” and systematically models the pipeline of feature-complexity–driven uncertainty quantification, dynamic confidence adjustment, logic-based fallback, and PAAE-based spatial enhancement. Therefore, even without adding extra experiments, we believe LRConfNet makes an independent contribution to the more specific problem of hierarchical confidence control and path optimization under degradation. We will continue to track and incorporate the most recent related research. (Please see Section 2.2 in the revised manuscript for the concrete updates.)

---

### Meta-Review · Area_Chair_z6aD · 2026-01-07

**Summary:**

The paper introduces LRConfNet, a framework for hierarchical classification of degraded images that integrates uncertainty quantification  and logical reasoning regularization. The primary contribution is a confidence-aware mechanism that allows the model to revert to coarse-grained predictions when fine-grained uncertainty is high, ensuring better reliability in noisy or blurred conditions. While the reviewers found the concept of logical consistency in confidence estimation interesting, they also pointed out sever concerns including that the paper presents as a "heavy stack" of technical tricks with limited novelty, insufficient theoretical justification, and a need of better presentation quality.

**Reviewer Concerns:**

Several primary concerns persist regarding the manuscript: (1) the framework is perceived as an overly complex "pile-up" of modules (e.g., UQ, attention enhancement, and dynamic thresholding) that lacks a focused narrative or a singular, clear innovative core; (2) the evaluation remains limited to specific datasets (remote sensing and birds), leaving the method's utility in broader domains, such as medical imaging, unproven; and (3) presentation quality (such as squeezed text, inconsistent table formatting, and overly complex figures) hinder readability and needs improvement.

While the authors provided additional discussions regarding hyperparameters and updated Figures 1 and 2 in the revision, the core concerns about a heavy stack of technical tricks with diluted contributions remain and the empirical evaluation remains limited to narrow domains.

**Reviewer Scores:**

Reviewer BoTe is likely to maintain a score of 6. However, the other three reviewers may maintain their scores of 2, 4, and 4 due to persistent concerns regarding the framework's reliance on a 'heavy stack' of technical tricks and the limited generalization of the evaluation to other domains.

---

### Decision · Program_Chairs · 2026-01-26

Reject